# Stochastic Chaos and Markov Blankets

**DOI:** 10.3390/e23091220

**Published:** 2021-09-17

**Authors:** Karl Friston, Conor Heins, Kai Ueltzhöffer, Lancelot Da Costa, Thomas Parr

**Affiliations:** 1Wellcome Centre for Human Neuroimaging, Institute of Neurology, University College London, London WC1N 3AR, UK; k.friston@ucl.ac.uk (K.F.); kai.ueltzhoeffer@med.uni-heidelberg.de (K.U.); l.da-costa@imperial.ac.uk (L.D.C.); thomas.parr.12@ucl.ac.uk (T.P.); 2Department of Collective Behaviour, Max Planck Institute of Animal Behavior, 78457 Konstanz, Germany; 3Centre for the Advanced Study of Collective Behaviour, 78457 Konstanz, Germany; 4Department of Biology, University of Konstanz, 78457 Konstanz, Germany; 5Department of General Psychiatry, Centre of Psychosocial Medicine, Heidelberg University, Voßstraße 2, 69115 Heidelberg, Germany; 6Department of Mathematics, Imperial College London, London SW7 2AZ, UK

**Keywords:** thermodynamics, information geometry, variational inference, Bayesian, Markov blanket

## Abstract

In this treatment of random dynamical systems, we consider the existence—and identification—of conditional independencies at nonequilibrium steady-state. These independencies underwrite a particular partition of states, in which internal states are statistically secluded from external states by blanket states. The existence of such partitions has interesting implications for the information geometry of internal states. In brief, this geometry can be read as a physics of sentience, where internal states look as if they are inferring external states. However, the existence of such partitions—and the functional form of the underlying densities—have yet to be established. Here, using the Lorenz system as the basis of stochastic chaos, we leverage the Helmholtz decomposition—and polynomial expansions—to parameterise the steady-state density in terms of surprisal or self-information. We then show how Markov blankets can be identified—using the accompanying Hessian—to characterise the coupling between internal and external states in terms of a generalised synchrony or synchronisation of chaos. We conclude by suggesting that this kind of synchronisation may provide a mathematical basis for an elemental form of (autonomous or active) sentience in biology.

## 1. Introduction

The physics of far from equilibrium or nonequilibrium systems represents a current focus of much theoretical research; especially at the interface between the physical and life sciences [1,2,3,4,5,6,7,8,9,10,11,12]. One example of this is the so-called free energy principle that attempts a formal account of sentient behaviour based upon the properties that random dynamical systems with a nonequilibrium steady-state density must possess [13,14]. In brief, the free energy principle is a variational principle of stationary action applied to a particular partition of states, where this partition rests upon conditional independencies [15]. Specifically, if the states of a system, whose dynamics can be described with random or stochastic differential equations (e.g., the Langevin equation), possess a Markov blanket, then an interesting interpretation of their dynamics emerges: the conditional independence in question means that a set of (internal) states are independent of another (external) set, when conditioned upon blanket states. The internal states can then be cast as representing, in a probabilistic fashion, external states. From this, one can elaborate a physics of sentience or Bayesian mechanics that would be recognised in theoretical neuroscience and biology [13,16]

However, the existence—and functional form—of the probability densities that underwrite the particular partition above is generally assumed without proof or demonstration. In this work, we attempt an end-to-end derivation of a Markov blanket, starting with a normal form of stochastic chaos and ending with the functional form of a synchronisation map. This (conditional synchronisation) mapping embodies the generalised synchrony between internal and external states, when conditioned on blanket states.

However, to do this we have to solve another problem first. A Markov blanket is defined in terms of conditional dependencies entailed by the joint density over some states; in this instance, the (steady-state) solution to the density dynamics of a random dynamical system. The problem here is that a random dynamical system is specified by random or stochastic equations of motion. This means we require a procedure to define the steady-state density from the functional form of the equations of motion. In other words, we seek a procedure that returns the nonequilibrium steady-state density, given some equations of motion (and the statistics of random fluctuations).

The first section of this technical note describes one procedure using the Lorenz system [17,18] as the basis of a random dynamical system that exhibits stochastic chaos. This system possesses a pullback attractor that is itself a random variable [19,20,21]. In other words, there exists a probability density over the states in the long-term future that we will refer to as a nonequilibrium steady-state density. Technically, nonequilibrium in this context refers to the time irreversibility of the dynamics (also known as loss of detailed balance) that inherits from solenoidal or conservative flow of systemic states [1,22,23]. It is this conservative component of flow that underwrites stochastic chaos as nicely described in [24] and quantified by positive Lyapunov exponents [25], i.e., exponential divergence of trajectories and associated sensitivity to initial conditions [17]. We emphasise that stochastic chaos is an attribute of the flow, as opposed to the noise, in a stochastic differential equation.

In brief, the approach taken in the first section is to apply the Helmholtz decomposition to the solution of the Fokker Planck equation describing the density dynamics of any random dynamical system. The Helmholtz decomposition enables one to express the expected flow as the product of a flow operator and the gradients of a scalar field that corresponds to self-information (in information theory) or a potential function (in stochastic dynamics) [23,26,27,28,29]. By parameterising the components of the flow operator and potential using a polynomial expansion, we have a generic functional form for the equations of motion that entail the steady-state density implied by the potential. We approximate the flow with a quadratic expansion, which means the steady-state density reduces to a Gaussian form. We can treat this as a *Laplace approximation* to steady-state dynamics, which we note may differ from the steady-state of the original (non-approximated) system. Under this Laplace approximation, the Hessian or curvature of the potential corresponds to the inverse covariance or precision of the steady-state density, from which we can read off various statistical dependencies. We will see that for a Lorenz system, subject to appropriate random fluctuations, the third state is independent of the first two. This means that the third state has no blanket states and therefore there is no particular partition of interest.

Things get more interesting in the second section when we couple two Lorenz systems via their first states. We now identify conditional independencies in the Hessian of the Laplace approximation, which features a sparse structure and a number of Markov blankets. It then becomes possible to distinguish between internal and external states and see what happens when they are conditioned upon a blanket state. This conditioning gives rise to a (conditional) synchronisation map or manifold that supports generalised synchrony between the internal and external states.

The existence of this manifold means that one can provide a stipulative definition of a conditional distribution over external states that is parameterised by the conditional expectation of internal states. In other words, there exists a diffeomorphic map between expected internal states and (the sufficient statistics of) Bayesian (i.e., probabilistic) beliefs *about* external states. In the worked example—based upon a Laplace approximation—this manifold has a remarkably simple (linear) form. The ensuing conditional synchronisation licenses a description of dissipative flow at nonequilibrium steady-state in terms of a gradient flow on a variational free energy. This variational free energy replaces the self-information of a blanket state with a functional of the conditional distribution over external states. This substitution lends the gradient flow an interpretation in terms of inference or [30,31,32], more poetically, self-evidencing [33,34].

We conclude with a discussion of the implications and generalisations of the following (heuristic) proof of principle for understanding, simulating and characterising an elemental form of sentience in systems that self-organise to nonequilibria.

## 2. From Dynamics to Densities

The aim of this section is to get from the specification of a random dynamical system—in terms of its equations of motion—to the probability density over its states in the long-term future, from any initial conditions. We are only interested in systems that have a limit set; in other words, systems that possess an attractor. We want to describe systems that do not necessarily visit all possible states; are not necessarily time reversible; are stochastic in nature and potentially chaotic. In short, we are interested in stochastic chaos in systems with a pullback attractor [19,20]. Such systems can be described with stochastic differential equations, such as the Langevin equation describing the rate of change of states *x*, in terms of their flow *f*(*x*), and fast fluctuations *ω*. The fluctuations are usually assumed to be a Wiener process, i.e., normally distributed and temporally uncorrelated, with an amplitude of 2Γ (a diagonal covariance matrix):(1)$$\begin{array}{cc} \dot{x} & =f(x)+\omega \\ & \Rightarrow \\ \dot{p}(x) & =\nabla \cdot (\Gamma \nabla -f)p(x) \end{array}$$

The second quality is the Fokker–Planck equation [35], which provides an equivalent description of the stochastic process in terms of deterministic density dynamics. The dot notation indicates a partial temporal derivative. In the absence of random fluctuations (i.e., Γ = 0) the attractor corresponds to a limit set; namely, the set of states onto which the solutions of (1) converge. When the flow shows exponential divergence of trajectories, with positive Lyapunov exponents (i.e., real positive eigenvalues in the Jacobian of the flow), the system can be said to be chaotic [36,37]. If we reinstate the random fluctuations, the otherwise strange attractor or limit set is now replaced by a pullback attractor that accommodates the fact that every solution is a random variable [20]. In this case, the flow becomes the expected motion. If this flow shows exponential divergence of trajectories, we can impute stochastic chaos.

Figure 1 provides an example of stochastic chaos based upon the Lorenz system of equations. The solid line represents a solution based upon the flow (i.e., in the absence of random fluctuations), while the dotted line is a realisation of a random trajectory using random fluctuations with a variance of 16—and the following equations of motion:(2)$$\;f\left(x\right)=\left[\begin{array}{c} \sigma x_{2}-\sigma \;x_{1}\\ \;\rho x_{1}-x_{2}-x_{1}\;x_{3}\\ x_{1}\;x_{2}-\beta x_{3} \end{array}\right],\;J=\nabla f=\left[\begin{array}{ccc} -\sigma & \sigma & 0\\ \rho -x_{3} & -1 & -x_{1}\\ x_{2} & x_{1} & -\beta \end{array}\right]$$
with σ=10, $\beta =\frac{8}{3}$, and ρ=32. Note that the left-hand size of Equation (2) details the deterministic equations of motion of the Lorenz system, whereas the right-hand side is the Jacobian, i.e., the gradient of these equations with respect to the states. The Lorenz system will be the focus of what follows because it represents a canonical form of deterministic chaos. Originally, the Lorenz equations were derived from an Oberbeck–Boussinesq approximation to the equations describing Rayleigh–Bénard convection [38]. The ensuing equations were then subject to a spectral Galerkin approximation, to produce a set of three coupled, nonlinear ordinary differential equations [18]. This system has been studied extensively and is used to model many systems in the physical and life sciences, e.g., [39,40].

## 3. The Helmholtz Decomposition

From our perspective, we could regard the deterministic Lorenz system as describing the expected flow of a random dynamical system that is subject to random fluctuations. The flow of such systems—which possess a nonequilibrium steady-state density—can be expressed using a generalisation of the Helmholtz decomposition into dissipative (irrotational, curl-free) and conservative (rotational, divergence-free) components, where the latter are referred to as solenoidal flow [7,14,26,27,29,41,42,43]. For an introduction to this generalisation of the Helmholtz decomposition see Appendix A of [16].
(3)$$\begin{array}{cc} \dot{p}(x) & =0\Leftrightarrow \\ f(x) & =\underset{Solenoidal}{\underset{︸}{Q\nabla ℑ(x)}}-\underset{Gradient}{\underset{︸}{\Gamma \nabla ℑ(x)}}-\Lambda (x)=\underset{Flow}{\underset{︸}{\Omega \nabla ℑ(x)-\Lambda (x)}}\\ ℑ(x) & =-\ln p(x)\\ \Omega (x) & =Q(x)-\Gamma (x)\\ Q(x) & =-Q{\left(x\right)}^{T}\Rightarrow \nabla \cdot Q\nabla ℑ=0\\ \Lambda {\left(x\right)}_{i} & =\sum_{j}\frac{\partial \Omega_{ij}}{\partial x_{j}} \end{array}$$

This decomposition can be understood intuitively as a decomposition of the expected flow into two orthogonal parts. The first (dissipative) part performs a Riemannian gradient descent on the negative logarithm of the steady-state density [44,45] which can be interpreted as the self-information ℑ(x)=−lnp(x) of any given state or as some potential function [7,29,46]. The second part of the flow is the solenoidal circulation on the isocontours of the steady-state density. This component breaks detailed balance and renders the steady-state a *nonequilibrium* steady-state. The third term, Λ(x), can be regarded as a correction or housekeeping term that accounts for changes in the flow operator Ω(x), over state-space, i.e., changes in the amplitude of random fluctuations or solenoidal flow.

Notice that the gradient flow depends upon the amplitude of the random fluctuations. Intuitively, this means that at nonequilibrium steady-state, the dispersive or dissipating effect of random fluctuations is balanced by the gradient flows up the log density gradients. This means that as the amplitude of random fluctuations decreases, the rotational flow comes to dominate leading, ultimately, to classical (e.g., Lagrangian) motion of massive bodies with periodic orbits or the deterministic chaos of chaotic attractors.

### 3.1. Functional Forms

Our objective is to identify the functional form of the self-information or potential function that describes the nonequilibrium steady-state density. If the solenoidal flow were not state-dependent, this would be a relatively straightforward exercise. This follows because there are linear constraints on the rotational flow that inherit from its skew symmetric, form, i.e., $Q=-Q^{T}$, and the fact that the Hessian of the potential has to be symmetric, i.e., $\nabla^{2}ℑ=H=H^{T}$. In brief, given the amplitude of random fluctuations and the equations of motion—and implicitly the Jacobian J=∇f—the rotational operator is given by the following (using “/” and “\” to denote right and left matrix division, respectively):(4)$$\begin{array}{cc} f(x) & =\Omega \nabla ℑ(x)\\ & \Rightarrow \nabla f=J=\Omega \nabla^{2}ℑ=\Omega H\Rightarrow \Omega \backslash J=H=J^{T}/\Omega^{T}\\ & \Rightarrow J\Omega^{T}=\Omega J^{T}\Rightarrow J{\left(Q-\Gamma \right)}^{T}=(Q-\Gamma )J^{T}\\ & \Rightarrow JQ+QJ^{T}=\Gamma J^{T}-J\Gamma \\ & \Rightarrow vec(Q)=(I⊗J+J⊗I)\backslash vec(\Gamma J^{T}-J\Gamma ) \end{array}$$

However, in general, we cannot discount the correction term that arises when flow operators are a function of the states. Indeed, it is this state dependency that underwrites stochastic chaos. This presents us with a more difficult problem. This problem can be finessed by using polynomial expansions of the flow operator and the potential as follows, for *n* states up to polynomial order *m*:(5)$$\begin{array}{cc} f(x) & =\Omega (x)\nabla ℑ(x)-\Lambda (x)\\ ℑ(x) & =x\cdot h\\ \Omega (x) & =\left[\begin{array}{ccc} x\cdot q_{11}-\Gamma_{1} & \cdots & x\cdot q_{1n}\\ \vdots & \ddots & \vdots \\ -x\cdot q_{1n} & \cdots & x\cdot q_{nn}-\Gamma_{n} \end{array}\right]\\ x & =x_{1}⊗x_{2}⊗\ldots ⊗x_{n},\;x_{i}=[1,x_{i},\frac{1}{2}x_{i}^{2},\ldots ,\frac{1}{m!}x_{i}^{m}]\\ h & =[h_{1},h_{2},\ldots ]\\ q & =[q_{11},q_{12},\ldots ,q_{nn}]=[q_{1},q_{2},\ldots ] \end{array}$$

Note, that this parameterisation allows for state-dependent changes in the amplitude of random fluctuations, encoded by the leading diagonal of the flow operator. The leading diagonal of the flow operator encodes the (negative) amplitude of the random fluctuations for each dimension in state-space. This follows from the definition of the skew-symmetric solenoidal matrix (with an all-zero diagonal). This also assumes spatially uncorrelated noise, i.e., diagonal Γ. With this functional form, it is straightforward to solve for the polynomial coefficients of the flow operator by solving the following simultaneous equations for a series of (sample) points: x∈ℝn.
(6)∂ε∂q=0, ∂ε∂h=0, f(x)=f(x,q,h)+∂ε∂q⋅qε(x)=f(x)−f(x,q,h)f(x,q,h)=Ω(x,q)∇x⋅h−Λ(x,q)∂ε∂qij=∂Λ∂qij−∂Ω∂qij∇x⋅h ∂Λi∂qij={∇jxi≤j−∇jxi>j,∂Ωij∂qij=−∂Ωji∂qij=x,∂ε∂h=Ω(x,q)∇x

Here, ε(x) can be regarded as the difference between a target flow f(x) and the flow predicted by the polynomial parameters f(x,q,h) at each location in state-space x. The solution to (6) minimises this difference—in a least squares sense. Practically, solving (6) involves selecting a few sample points in state-space and iterating
(7)$$\begin{array}{c} q\leftarrow q+{\left(\frac{\partial \varepsilon }{\partial q}\right)}^{-1}\varepsilon \\ h\leftarrow h+{\left(\frac{\partial \varepsilon }{\partial h}\right)}^{-1}\varepsilon \end{array}$$
until convergence. This means we have to specify the order of the polynomial expansion—and which sample points to use. Inspection of the Lorenz system suggests that a second-order polynomial approximation is sufficient, given the flow is second order in the states. This ansatz has an interesting implication: if the self-information can be approximated with a second-order polynomial, it means the nonequilibrium steady-state is approximately Gaussian. This is known as the Laplace approximation in statistics [47,48]. Here, we generalise the notion of a Laplace approximation to cover not just the quadratic form of the log density but also the solenoidal flow that underwrites nonequilibrium dynamics.

### 3.2. The Lorenz System Revisited

If the polynomial expansion is up to second-order, then it is sufficient to select three or more sample points along any dimension to render the simultaneous equations in (6) fully determined. If we allow the amplitude of random fluctuations to take positive or negative values, solving (6) for any set of sample points furnishes an exact solution for the Lorenz system, in which most of the polynomial coefficients vanish:


(8)
$$\begin{array}{cc} ℑ & =h_{5}\;x_{1}\;x_{2}+h_{7}\;x_{3}\\ f & =\Omega \nabla ℑ-\Lambda \\ & =\left[\begin{array}{ccc} -\Gamma_{1} & q_{11} & 0\\ -q_{11} & q_{37}\;x_{3}-\;\Gamma_{2} & q_{44}\;x_{2}\\ 0 & -q_{44}\;x_{2} & q_{57}\;x_{3}+q_{55}\;x_{1}\;x_{2}-\Gamma_{3} \end{array}\right]\left[\begin{array}{c} h_{5}\;x_{2}\\ h_{5}\;x_{1}\\ h_{7} \end{array}\right]-\left[\begin{array}{c} 0\\ 0\\ q_{44}-q_{57} \end{array}\right]\\ & =\left[\begin{array}{c} -\Gamma_{1}h_{5}\;x_{2}+h_{5}\;q_{11}\;x_{1}\\ -\Gamma_{2}h_{5}\;x_{1}-\left(h_{5}\;q_{11}-h_{7}q_{44}\;\right)\;x_{2}+h_{5}\;q_{37}\;x_{1}x_{3}\\ q_{57}-q_{44}-\Gamma_{3}h_{7}+h_{7}q_{57}\;x_{3}-\left(h_{5}q_{44}-h_{7}q_{55}\right)x_{1}x_{2} \end{array}\right]=\left[\begin{array}{c} \sigma \;x_{2}-\sigma x_{1}\\ \rho x_{1}-x_{2}-x_{1}\;x_{3}\\ x_{1}\;x_{2}-\beta x_{3} \end{array}\right]\Rightarrow \{\begin{array}{c} \sigma =-h_{5}\;q_{11}\\ \beta =\;-h_{7}q_{57}\\ \rho =-h_{5}\;\Gamma_{2} \end{array} \end{array}$$


This can be regarded as a generalised Helmholtz decomposition of the Lorenz system. This decomposition has a remarkably simple potential that includes the product of the first and second states—and is first-order in the third state. Figure 1 (right middle panel) shows a solution to the above Laplace version, which is indistinguishable from the Lorenz system. The lower left panel shows the potential as a function of time, while the lower right panel plots the path of the first and second states over the potential, shown in image format. One could picture the Lorenz system as performing a gradient flow on the potential, which is high when the first and second states take large values, with the same sign. However, this gradient flow is complemented with a solenoidal flow and a (constant) correction or housekeeping flow in the third dimension. These three components are shown as a quiver plot in the upper panel of Figure 2.

The ensuing decomposition of the Lorenz system allows us to express the parameters of the Lorenz system in terms of the coefficients of the Laplace form, as in the last set of qualities in (8). It is also possible to express the polynomial coefficients as functions of the Lorenz parameters using recursive elimination (as summarised in the Appendix A).

Although it is interesting to decompose the flow of the Lorenz system in this fashion, it does not really help in terms of identifying the steady-state density of a random dynamical system. This is because the Hessian associated with (8) is not positive definite and does not admit an interpretation in terms of a proper probability density. One might intuit this by noting that the log steady-state density is an odd polynomial in $x_{3}$ and so is incompatible with a proper density. In addition, the amplitude of random fluctuations is negative in some domains of state-space, implying complex fluctuations. Indeed, the equations of motion we have written down are deterministic, implying that there is no stochasticity in the sample paths of this process (assuming a fixed initial condition). This speaks to the fact that the Helmholtz decomposition of the deterministic Lorenz system is not a description of a dynamical system with random fluctuations, i.e., systems in which the dissipative part of the flow operator is positive definite. We therefore need to look beyond the Lorenz attractor.

### 3.3. Beyond the Lorenz System

Figure 3 shows a Laplace approximation to the Lorenz system using 4^3^ = 64 sample points on a hypergrid spanning ±8, centred on [0, 0, 28]. This solution of (6) fixed the variance of random fluctuations at 2Γ = [1/8, 1/16, 1/32] by setting the leading diagonal polynomial coefficients to $q_{ii}=0$. The upper panels show a solution based upon the expected flow, from three directions, overlaid on the marginal steady-state density. This density is determined by the polynomial coefficients of the self-information, according to (4), with the following functional form:


(9)
$$\begin{array}{cc} p(x) & =\frac{1}{Z}e^{-ℑ}=\frac{1}{Z}e^{-x\cdot h}\\ ℑ & =h_{7}x_{3}\;+h_{5}\;x_{1}\;x_{2}+h_{3}\frac{1}{2}x_{3}\;x_{1}^{2}+\frac{1}{2}h_{6}\;x_{2}^{2}+\frac{1}{2}h_{10}\;x_{3}^{2} \end{array}$$


The lower panels show the trajectory of the three states as a function of time and plotted against each other to reveal the shape of the underlying attractor. The solid lines show a solution in the absence of random fluctuations (i.e., the expected flow). The dotted lines show a solution to the stochastic differential equation afforded by the Laplace approximation:


(10)
$$\begin{array}{cc} f & =\Omega \nabla ℑ-\Lambda \\ & =\left[\begin{array}{ccc} -\Gamma_{1} & Q_{12} & 0\\ -Q_{12} & -\Gamma_{2} & Q_{23}\\ 0 & -Q_{23} & -\Gamma_{3} \end{array}\right]\left[\begin{array}{c} h_{3}\;x_{1}+h_{5}\;x_{2}\\ h_{5}\;x_{1}+h_{6}\;x_{2}\\ h_{7}+h_{10}\;x_{3} \end{array}\right]-\left[\begin{array}{c} -q_{15}\;x_{1}-q_{16}\;x_{2}\\ q_{13}\;x_{1}+q_{15}\;x_{2}-q_{49}\;x_{2}\\ q_{44}+q_{49}\;x_{3} \end{array}\right]\\ Q_{12} & =q_{11}+q_{15}\;x_{1}\;x_{2}+\frac{1}{2}q_{13}\;x_{1}{}^{2}+\frac{1}{2}q_{16}\;x_{2}{}^{2}\\ Q_{23} & =q_{42}\;x_{1}+q_{44}\;x_{2}+q_{49}\;x_{2}\;x_{3} \end{array}$$


Note that this is not a stochastic Lorenz system because the expected flows of the Laplace approximation are not the Lorenz flows, i.e., the Laplace system approximates the Lorenz system, under the constraint that the amplitude of random fluctuations take a particular value. This is shown in the lower panel where there are differences between the Laplace and Lorenz flows in all three dimensions. However, the Laplace system inherits some aspects of the Lorenz attractor, such as the “butterfly wings” in the upper right panel—although one might argue the attractor is more exotic or itinerant than the Lorenz attractor. On the other hand, the gradient flows now have a much simpler structure because they are performing a gradient descent on a quadratic potential (or a negative log-Gaussian steady-state density). This can be seen by returning to Figure 2 (lower panel), which decomposes the flow into gradient, solenoidal and correction components. In contrast to the Lorenz flow, the expected flow of this Laplace system (blue arrows) is towards the maximum of the steady-state density.

Clearly, there will be a different Laplace approximation for each set of sample points (and, indeed, choice of Γ). We have chosen a system whose expected flow has a limit set with a similar topology to that of the Lorenz attractor. This can be established by estimating the expected Lyapunov exponents λ(x)=eig(J) under the steady-state density
(11)$$\begin{array}{cc} \lambda_{i} & =\int p(x)\lambda {\left(x\right)}_{i}dx\\ d_{H} & \approx j+\frac{\sum_{i=1}^{j}\lambda_{i}}{\left|\lambda_{j+1}\right|},\;\mathrm{where}\;\sum_{i=1}^{j}\lambda_{i}⩾0\;\mathrm{and}\;\lambda_{i}\geq \lambda_{i-1} \end{array}$$

One can then estimate the Lyapunov dimension (by the Kaplan-Yorke conjecture, the Lyapunov dimension approximates the Hausdorff dimension [49]) which, in this example, was 2.48, (with eigenvalues: 0.0766, 0.0056, −0.1713), when estimated on a 16^3^ hypergrid spanning ±32, centred on [0, 0, 28]. This compares with the analytic Lyapunov dimension of the Lorenz system for the parameters we have used [50]:(12)$$d_{H}=3-\frac{2(\sigma +\beta +1)}{\sigma +1+\sqrt{{\left(\sigma -1\right)}^{2}+4\sigma \rho }}=2.43$$

In short, the flows of the Lorenz system and its Laplace approximation have an attracting set with between two and three dimensions. For our purposes, the Laplace approximation is easier to handle than the Lorenz system because the functional forms of the flow and potential are immediately at hand. Figure 4 leverages the implicit access to the potential function showing that—at nonequilibrium steady-state—the first two states are highly correlated, while the third state is statistically independent of the first pair. This is a consequence of the flow constraints implied by the directed influence of the first state on the third. We will see why this is the case below.

The upper panels of Figure 4 show the Jacobian, covariance and Hessian, displayed as the log of their absolute values. This enables their sparsity structure (i.e., zero elements) to be identified easily. The Jacobian encodes the dynamical coupling—in other words, which states influence the flow of other states. For the Lorenz system, the third state does not influence the first state, which means the corresponding entry in the Jacobian is zero (using *u* and *v* to denote generic state indices, and ~ to denote the complement):(13)$$\begin{array}{cc} J_{uv} & =\frac{\partial f_{u}}{\partial x_{v}}=0:\forall x\Leftrightarrow f{\left(x\right)}_{u}=f{\left(x_{\tilde{v}}\right)}_{u}\\ x & =(x_{\tilde{v}},x_{v}) \end{array}$$

In short, we can read the Jacobian as encoding sparsity constraints on dynamical coupling, namely the absence of an influence of one state on the dynamics of another. In a similar vein, the Hessian encodes the conditional dependencies as follows.

The steady-state covariance is the inverse of the Hessian, which—in this example—shows that the covariance between the third state and the first two states is zero. More tellingly, the corresponding entries in the Hessian are zero. Zero Hessian elements are particularly revealing because they imply conditional independence.
(14)$$\begin{array}{cc} H & =\left[\begin{array}{ccc} h_{3} & h_{5} & 0\\ h_{5} & h_{6} & 0\\ 0 & 0 & h_{10} \end{array}\right]=\Sigma^{-1}=\nabla^{2}ℑ\\ H{\left(x\right)}_{uv} & =\frac{\partial^{2}ℑ}{\partial x_{v}\partial x_{u}}=0:\forall x\\ & \Leftrightarrow ℑ(x_{u}|b,x_{v})=ℑ(x_{u}|b)\\ & \Leftrightarrow ℑ(x_{u},x_{v}|b)=ℑ(x_{u}|b)+ℑ(x_{v}|b)\\ & \Leftrightarrow (x_{u}⊥x_{v})|b:b=x_{\tilde{u},\tilde{v}} \end{array}$$

The second equality in (14) holds for any nonequilibrium steady-state density. In other words, if the second derivative or Hessian of the log density—with respect to two states—is zero (almost) everywhere, then those two states are conditionally independent, given the remaining states. Intuitively, this means that changing one state does not change the conditional density over the other. If this is true at every point in state-space, the two states are conditionally independent. The advantage of working with a Laplace approximation is that the Hessian is the same everywhere. This means that if the Hessian is zero at one point, it is zero everywhere and conditional independence is assured. Note that the Laplace form is not necessary for conditional independence—it just makes it easier to establish because the functional form of the nonequilibrium steady-state density is known. The lack of statistical dependence between one conditional density and the other implies—and is implied by—a corresponding zero entry in the Hessian matrix. This follows straightforwardly from the definition of conditional independence in the case of a Laplace form (i.e., a multivariate Gaussian density), which can be expressed using the inverse covariance, i.e., the Hessian [51].

In other words, if two states are conditionally independent, given the remaining (blanket) states b, then their corresponding entries in the Hessian are zero—and vice versa. In this instance, the third state is conditionally independent of the first and second states, which means it has no blanket states. Slices through the nonequilibrium steady-state density—in the middle panels of Figure 4—reaffirm the independence between the first pair of states and the third state. The only dependencies at nonequilibrium steady-state are between the first and second, shown in the bottom row. These are further illustrated in terms of the conditional density of the first state, given the second, in the lower panel, expressed as a function of time. The conditional mean and covariance are denoted by the white lines and shaded areas, respectively. The red line corresponds to the stochastic trajectory of the first state from the Figure 3.

### 3.4. Beyond the Laplace System

The preceding treatment leverages the simplicity of the Laplace approximation to stochastic chaos, in which sparsity constraints on the Hessian are easy to identify or implement. One might ask whether more accurate approximations to any given system are obtained when increasing the order of the polynomial expansion. Figure 5 illustrates a high-order approximation to a Lorenz system that provides a more accurate approximation to the Lorenz system, in terms of the expected flow. However, the ensuing steady-state density is not Gaussian because the Hessian is now state-dependent. This follows because the Hessian is the double derivative of the potential with respect to the states, which is only constant in the absence of third and higher order terms.

Care has to be taken when fitting the desired (or observed) flow with high-order polynomial expansions because, effectively, this fitting is a model inversion problem—where the forward or generative model is supplied by the polynomial form of the Helmholtz decomposition. Because this sort of generative model is highly nonlinear, its inversion can easily encounter local minima. This problem can be finessed by replacing the least-squares scheme in (6) with a standard *maximum a posteriori* scheme; here, variational Laplace [47]. In addition, one can constrain the polynomial expansion to ensure (to first-order) that the Hessian is positive definite everywhere. The example in Figure 5 used the following parameterisation, in which the Hessian was parameterised with the product of two symmetric (kernel) matrices:(15)$$\begin{array}{cc} ℑ(x) & =\frac{1}{2}{\left(x-m\right)}^{T}K^{T}K(x-m)\\ m_{i} & =m_{i},K_{ij}=K_{ji}=x\cdot k_{ij} \end{array}$$

This allows the Hessian to change over state-space, while remaining positive definite everywhere. This means that the steady-state density over one or two states is only Gaussian when conditioned upon the remaining states. The example in Figure 5 used polynomial expansions to second-order for the elements of the solenoidal operator and first-order for the elements of the kernel matrix. Effectively, this means the Hessian contains fourth order terms; however, its parameterisation in (15) ensures it is positive definite everywhere. These constraints may be useful for fitting dynamics or density learning in an empirical setting, where high-order polynomial expansions may provide a more expressive generative model. However, as we will see below, the Laplace approximation is more suitable for our purposes.

### 3.5. Summary

In summary, this section has shown that it is fairly straightforward to construct a random dynamical system with stochastic chaos, using the Helmholtz decomposition and polynomial expansions. Second-order or quadratic approximations can be regarded as a dynamical form of the Laplace approximation. Under these approximations, there is a remarkably simple (Gaussian) form for the nonequilibrium steady-state density. This form permits identification of conditional independencies via the Hessian (i.e., the inverse covariance or precision of the nonequilibrium steady-state density). As in deterministic systems, the chaotic aspect of the (expected) flow depends upon solenoidal couplings that underwrite stochastic chaos, with a fractional Lyapunov (Hausdorff) dimension.

It is important not to conflate the simplicity of a nonequilibrium steady-state density with the complexity of the underlying density dynamics. In other words, when prepared (or observed) in some initial state, the probability density can evolve in a complicated and itinerant fashion on various submanifolds of the pullback attractor. This kind of dynamical complexity is probably best measured with the information length, as the initial density evolves towards the steady-state density [52,53,54]. Effectively, the information length scores the number of probabilistic configurations the system occupies as it evolves over time [55].

Figure 6 illustrates this kind of itinerancy with the time-dependent density over a number of epochs, starting from an initial density around [4, 4, 8] in state-space. The marginal densities in Figure 6 show that the states that are most likely to be visited in the future follow a circuitous path—due largely to solenoidal flow—moving initially away from the mode of the steady-state density and then converging on it. Throughout, random fluctuations disperse the density, meaning the future becomes progressively uncertain. In short, being at a nonequilibrium steady-state does not preclude structured sojourns through state-space, it just means that ultimately the system will occupy a characteristic set of attracting states in the long-term future.

Technically, the example in Figure 6 involved solving the density dynamics in terms of a time-dependent surprisal, using the following reformulation of the Fokker–Planck equation in terms of a time-dependent potential, ${ℑ}_{\tau }=-\ln p_{\tau }$:(16)$$\begin{array}{cc} {\dot{p}}_{\tau } & =\nabla \cdot \Gamma \nabla p_{\tau }-p_{\tau }\nabla \cdot f(x)-f(x)\cdot \nabla p_{\tau }\Rightarrow \\ {\dot{ℑ}}_{\tau } & =(\nabla -\nabla {ℑ}_{\tau })\cdot \Gamma \nabla {ℑ}_{\tau }+\nabla \cdot f(x)-f(x)\cdot \nabla {ℑ}_{\tau }\\ {\dot{p}}_{\tau } & =-p_{\tau }{\dot{ℑ}}_{\tau },\nabla p_{\tau }=-p_{\tau }\nabla {ℑ}_{\tau },\nabla^{2}p_{\tau }=-p_{\tau }\nabla^{2}{ℑ}_{\tau }+p_{\tau }\nabla {ℑ}_{\tau }\nabla {ℑ}_{\tau } \end{array}$$

Because the flow at each point in state-space does not change with time, we can express the time-dependent surprisal above in terms of the steady-state potential of any Laplacian system as follows:(17)$$\begin{array}{cc} f(x) & =f(x,q,h)=(Q-\Gamma )\nabla ℑ-\Lambda \\ & \Rightarrow \\ {\dot{ℑ}}_{\tau } & =\underset{Dissipative}{\underset{︸}{\left(\nabla -\nabla {ℑ}_{\tau })\cdot \Gamma \nabla ({ℑ}_{\tau }-ℑ\right)}}-\underset{Solenoidal}{\underset{︸}{\nabla {ℑ}_{\tau }\cdot Q\nabla ℑ}}+\underset{Correction}{\underset{︸}{\nabla ({ℑ}_{\tau }-ℑ)\cdot \Lambda }} \end{array}$$

This shows that when the potential difference ${ℑ}_{\tau }-ℑ$ is zero everywhere, there is no further change in the surprisal, and the density converges to steady-state. The number of probabilistic configurations—or states visited prior to convergence—can now be scored with the information length using ${\dot{ℑ}}_{\tau }$, as described in the appendix.

When constructing a Laplace system that inherits the flow of a Lorentz system, we find that the nonequilibrium steady-state density has a simple dependency structure, in which one state is independent of the remaining states. In the next section, we turn to a more interesting set up, which features conditional independencies and the emergence of Markov blankets.

## 4. Markov Blankets and the Free Energy Principle

In this section, we repeat the analysis of the previous section but approximate two Lorenz systems that are coupled to each other through their respective first states. This induces a richer conditional independence structure, from which one can identify internal and external states that are independent when conditioned upon blanket states. The existence of this particular partition licences an interpretation of the ensuing generalised synchrony [37,56,57], in which the conditional expectation of internal states parameterises probabilistic or Bayesian beliefs *about* external states. This notion can be formulated in terms of a variational free energy functional that underwrites the free energy principle [13,15].

### 4.1. Sparsely Coupled Systems

The form of coupling was chosen to be as simple
and symmetric as possible (with a coupling parameter $\chi \;=-\frac{1}{2}$ )
(18)$$f=\left[\begin{array}{c} \sigma \;x_{2}-\sigma \left(\chi \;x_{4}-x_{1}\;\left(\chi -1\right)\right)\\ \rho x_{1}-x_{2}-x_{1}\;x_{3}\\ x_{1}\;x_{2}-\beta x_{3}\\ \sigma x_{5}-\sigma \;\left(\chi x_{1}-x_{4}\;\left(\chi -1\right)\right)\\ \rho x_{4}-x_{5}-x_{4}\;x_{6}\\ x_{4}\;x_{5}-\beta x_{6} \end{array}\right]\text{, }J={\left[\begin{array}{cccccc} \sigma \;\left(\chi \;-1\right) & \sigma \; & 0 & -\sigma \;\chi \; & 0 & 0\\ \rho -x_{3} & -1 & -x_{1} & 0 & 0 & 0\\ x_{2} & x_{1} & -\beta & 0 & 0 & 0\\ -\sigma \;\chi \; & 0 & 0 & \sigma \;\left(\chi \;-1\right) & \sigma & 0\\ 0 & 0 & 0 & \rho -x_{6} & -1 & -x_{4}\\ 0 & 0 & 0 & x_{5} & x_{4} & -\beta \end{array}\right]}^{'}$$

In brief, this means that the influences on the motion of the first state of both systems comprises a mixture of the first states of each. Another way of looking at this is that the flow of the first states is determined in the usual way by the states of each system in isolation but are supplemented with a prediction error, i.e., the difference between the homologous states of both systems.


(19)
$$\begin{array}{cc} \sigma (\chi \;x_{4}-x_{1}\left(\chi -1\right)) & =\sigma \left(x_{1}+\chi \;\varepsilon \right)\\ \varepsilon & =x_{4}-x_{1} \end{array}$$


This means that there exists a solution to the expected flow in which homologous states are (on average) identical. In this instance, the prediction error is (on average) zero and the joint system exhibits identical synchronisation of chaos. Another way of expressing this is that the joint system has an identical synchronisation map that constitutes a pullback attractor for the joint system. We are interested in the implications of this synchronisation map; particularly in the presence of a Markov blanket. In short, we are interested in the synchronisation between states that are conditionally independent—and whether this provides the mathematical basis for an elementary kind of inference or sentience.

The Laplace approximation to the joint system in (19) is summarised in Figure 7, using the same format as Figure 3. It can be seen that the solution based upon expected flows has a slightly more complicated form due to the coupling between the two systems; however, the form of the nonequilibrium steady-state density remains largely unchanged. In this figure, the upper panel shows the states of the first system, while the lower panel shows all six states. The functional form of the Helmholtz decomposition for these coupled systems is as follows:


(20)
$$\begin{array}{c} f=\Omega \nabla ℑ-\Lambda \\ \Omega =\left[\begin{array}{cccccc} -\Gamma_{1} & Q_{12} & 0 & Q_{14} & 0 & 0\\ -Q_{12} & -\Gamma_{2} & Q_{23} & 0 & 0 & 0\\ 0 & -Q_{23} & -\Gamma_{3} & 0 & 0 & 0\\ -Q_{14} & 0 & 0 & -\Gamma_{1} & Q_{45} & 0\\ 0 & 0 & 0 & -Q_{45} & -\Gamma_{2} & Q_{56}\\ 0 & 0 & 0 & 0 & -Q_{56} & -\Gamma_{3} \end{array}\right],\;\nabla^{2}ℑ=H=\left[\begin{array}{cccccc} h_{3} & h_{5} & 0 & h_{12} & 0 & 0\\ h_{5} & h_{6} & 0 & 0 & 0 & 0\\ 0 & 0 & h_{10} & 0 & 0 & 0\\ h_{12} & 0 & 0 & h_{15} & h_{20} & 0\\ 0 & 0 & 0 & h_{20} & h_{21} & 0\\ 0 & 0 & 0 & 0 & 0 & h_{28} \end{array}\right]\\ \Lambda =\left[\begin{array}{c} -q_{33}\;x_{1}-q_{34}\;x_{2}-q_{99}\;x_{4}\\ q_{31}\;x_{1}+q_{33}\;x_{2}-q_{204}\;x_{1}-q_{205}\;x_{2}\\ q_{200}+q_{205}\;x_{3}\\ q_{87}\;x_{1}-q_{468}\;x_{4}-q_{469}\;x_{5}\\ q_{463}\;x_{4}+q_{468}\;x_{5}-q_{558}\;x_{4}-q_{559}\;x_{5}\\ q_{548}+q_{559}\;x_{6} \end{array}\right] \end{array}$$


Figure 8 uses the same format as Figure 4 to illustrate the coupling in terms of the Jacobian and the steady-state density in terms of the covariance and Hessian. Notice that the dynamical coupling (Jacobian) now has a much sparser structure because the states of one system do not influence the states of the other, with the exception of the first states of each system (i.e., states one and four). Likewise, the Hessian has a sparse structure with several conditional independencies (i.e., zero entries). These conditional independencies can now be used to identify a particular partition.

### 4.2. Particular Partitions, Boundaries and Blankets

In associating stochastic equations of motion with a unique (nonequilibrium steady-state) probability density, we have a somewhat special set up, in which the influences entailed by the equations of motion place constraints on the conditional independencies in the steady-state density. These conditional independencies can be used to identify a particular partition of states into external states, sensory states, active states and internal states as follows. This is an important move in the sense that it enables us to separate states of a particle (i.e., internal states and their accompanying sensory and active states) from the remaining (i.e., external) states. In detail, the Helmholtz decomposition means that the Jacobian can be expressed in terms of the Hessian (with a slight abuse of the dot product notation):(21)$$\begin{array}{cc} f(x) & =\Omega \nabla ℑ-\Lambda \\ & \Rightarrow J=\Omega H+\nabla \Omega \cdot \nabla ℑ-\nabla \Lambda \\ & \Rightarrow J_{uv}=\frac{\partial f_{u}}{\partial x_{v}}=\sum_{i}\Omega_{ui}H_{iv}+\sum_{i}\frac{\partial \Omega_{ui}}{\partial x_{v}}\frac{\partial ℑ}{\partial x_{i}}-\sum_{i}\frac{\partial^{2}\Omega_{ui}}{\partial x_{i}\partial x_{v}} \end{array}$$
This expansion can be used to motivate a *sparse coupling conjecture*, where sparse coupling is defined as an absence of coupling between two states. Sparse coupling means that the Jacobian coupling between states *u* and *v* is zero:(22)$$\begin{array}{cc} J{\left(x\right)}_{uv} & =0\\ & \Rightarrow \sum_{i}\Omega {\left(x\right)}_{ui}H{\left(x\right)}_{iv}+\ldots =0\\ & \Rightarrow Q{\left(x\right)}_{uv}H{\left(x\right)}_{vv}-\Gamma {\left(x\right)}_{u}H{\left(x\right)}_{uv}+\ldots =0 \end{array}$$
This constraint can be satisfied in one of two ways: either (i) all the terms in the expansion are identically zero or (ii) two or more terms cancel. If terms cancel everywhere in state space, then they must have exactly the same functional form. One can conjecture that the number of such solutions is vanishingly small for sufficiently high dimensional and nonlinear systems. If we ignore the (edge) cases in which two or more terms exactly cancel, then one can assume all of the terms are identically zero, which means one of their factors must be zero. However, there are two terms with nonzero factors; namely, those involving $H{\left(x\right)}_{vv}$ and $\Gamma {\left(x\right)}_{u}$, which are nonzero (almost) everywhere. These terms are retained in the final expression in Equation (22). If they are identically zero, we have:(23)$$\begin{array}{cc} Q_{uv}H_{vv} & =0\Rightarrow Q_{uv}=-Q_{vu}=0\\ \Gamma_{u}H_{uv} & =0\Rightarrow H_{uv}=H_{vu}=0\Leftrightarrow (x_{u}⊥x_{v})|b \end{array}$$
This means that the corresponding elements of the solenoidal operator $Q_{vu}$ and Hessian $H_{uv}$ must be zero at every point in state space and the two states are conditionally independent, given the remainder.

Equation (23) only applies in the absence of edge cases. Although edge cases—where terms cancel—predominate in linear systems [58,59], the question is whether edge cases are rare or the norm for nonlinear systems. The higher-dimensional the system’s state-space is, the less likely it is that the terms in Equation (22) cancel. This is because the number of constraints (dot products equalling 0, c.f. the second line of Equation (22)) is supralinear in the number of states. However, clearly, to answer this question one has to commit to a functional form, of the sort offered by the Laplace approximation. Crucially, the sparse coupling conjecture holds for the Laplacian versions of both the Lorenz system and coupled Lorenz systems. One can check that the sparse coupling conjecture holds in the Helmholtz decomposition of the coupled systems in Equations (18) and (20). The conditional independence implicit in the Hessian in (20) can be verified using numerical solutions: for example, by showing that the partial correlation between two states converges to zero using solutions to the stochastic differential equations over increasing periods of time (see Figure 9). Note that sparse coupling is sufficient but not necessary for conditional independence in certain (nonlinear) random dynamical systems. For example, in (20) the second and third states are conditionally independent, despite the fact that they are reciprocally (and anti-symmetrically) coupled.

In short, the sparse coupling conjecture says that for sufficiently high-dimensional and nonlinear systems, any two states are conditionally independent *if one state does not influence the other*. This follows in a straightforward way from the fact that the covariance of random fluctuations and the Hessian are definite matrices with nonzero leading diagonals. This leads to the slightly counterintuitive conjecture that directed couplings between two states generally imply conditional independence. If one state influences another, how can they be conditionally independent? Perhaps the simplest intuition is to note that the fluctuations of any (wide sense stationary) state and its motion are independent. Heuristically, knowing the motion of something tells you nothing about where it is. In the present setting, if one state influences another, then the motion of the influenced state can be predicted. However, this provides no information about its location in state-space. 

The sparse coupling conjecture does not depend upon Gaussian assumptions about the nonequilibrium steady-state density. It implies that dynamical influence graphs with absent or directed edges admit a Markov blanket (which may or may not be empty). These independencies can be used to build a particular partition using the following rules, c.f. Figure 10:
The Markov boundary a⊂x of a set of internal states μ⊂x is the minimal set of states for which there exists a nonzero Hessian submatrix:$\exists H_{a\mu }\neq 0$. In other words, the internal states are independent of the remaining states, when conditioned upon their Markov boundary, which we will call *active states*. The combination of active and internal states will be referred to as *autonomous states*: α={a,μ};The Markov boundary s⊂x of autonomous states is the minimal set of states for which there exists a nonzero Hessian submatrix:$\exists H_{s\alpha }\neq 0$. In other words, the autonomous states are independent of the remaining states, when conditioned upon their Markov boundary, which we will call *sensory states*. The combination of sensory and autonomous states will be referred to as *particular states*: π={s,α}.The combination of active and sensory (i.e., boundary) states constitute *blanket states*: b={s,a};The remaining states constitute *external states*: x={η,s,a,μ}.

The names of active and sensory (i.e., blanket) states inherit from the literature, where these states are often associated with biotic systems that act on—and sense—their external milieu [13,60]. See Figure 10. In this setting, one can regard external states as influencing internal states via sensory states. This influence can be mediated vicariously through active states or directly, through directed influences, which do not affect the conditional independencies by Equation (23). The particular partition is usually expressed in terms of dynamical coupling under conditional independence assumptions, where sensory states are not influenced by internal states. Similarly, active states are not influenced by external states. The ensuing conditional independencies implied by a particular partition can be summarised as follows:(24)$$\begin{array}{cc} J_{\mu \eta }J_{\eta \mu }^{T} & =0\Rightarrow H_{\mu \eta }=0\Leftrightarrow (\mu ⊥\eta )|b\\ J_{a\eta }J_{\eta a}^{T} & =0\Rightarrow H_{a\eta }=0\Leftrightarrow (a⊥\eta )|s,\mu \\ J_{s\mu }J_{\mu s}^{T} & =0\Rightarrow H_{s\mu }=0\Leftrightarrow (s⊥\mu )|a,\eta \end{array}$$

From Equation (21), general forms for the flow and Jacobian of a particular partition are:(25)$$\begin{array}{cc} f(x) & =\Omega \nabla ℑ-\Lambda \\ \left[\begin{array}{c} f_{\eta }(x)\\ f_{s}(x)\\ f_{a}(x)\\ f_{\mu }(x) \end{array}\right] & =\left[\begin{array}{cccc} -\Gamma_{\eta } & Q_{\eta s} & & \\ -Q_{\eta s}^{T} & -\Gamma_{s} & Q_{sa} & \\ & -Q_{sa}^{T} & -\Gamma_{a} & Q_{a\mu }\\ & & -Q_{a\mu }^{T} & -\Gamma_{\mu } \end{array}\right]\left[\begin{array}{c} \nabla_{\eta }ℑ(\eta |b)\\ \nabla_{s}ℑ(b|\eta )\\ \nabla_{a}ℑ(b|\mu )\\ \nabla_{\mu }ℑ(\mu |b) \end{array}\right]-\Lambda \\ J(x) & =\Omega H+\nabla \Omega \cdot \nabla ℑ-\nabla \Lambda \\ & =\left[\begin{array}{cccc} -\Gamma_{\eta } & Q_{\eta s} & & \\ -Q_{\eta s}^{T} & -\Gamma_{s} & Q_{sa} & \\ & -Q_{sa}^{T} & -\Gamma_{a} & Q_{a\mu }\\ & & -Q_{a\mu }^{T} & -\Gamma_{\mu } \end{array}\right]\left[\begin{array}{cccc} H_{\eta \eta } & H_{\eta s} & & \\ H_{\eta s}^{T} & H_{ss} & H_{sa} & \\ & H_{sa}^{T} & H_{aa} & H_{a\mu }\\ & & H_{a\mu }^{T} & H_{\mu \mu } \end{array}\right]+\nabla \Omega \cdot \nabla ℑ-\nabla \Lambda \end{array}$$

In summary, active states are blanket states that are not influenced by external states, while sensory states are blanket states that are not influenced by internal states. Figure 10 illustrates this particular partition. Note that the edges in this graph represent the influences of one state on another, as opposed to a conditional dependency in a Bayesian graph. This is important because directed influences imply conditional independence. These conditional independencies are manifest as zero entries in the Hessian matrix, which defines the sparsity of conditional dependencies that inherit from the sparse (directed) coupling of the dynamics.

One might ask why a particular partition comprises four sets of states? In other words, why does a particular partition consider two Markov boundaries—sensory and active states? The reason is that the particular partition is the minimal partition that allows for directed coupling with blanket states. For example, sensory states can influence internal states—and active states can influence external states—without destroying the conditional independencies of the particular partition (these directed influences are illustrated in the upper panel of Figure 10 as dotted arrows).

Returning to our example of two coupled systems, we can now apply the above rules to identify the Markov blanket of internal states: for example, the last pair of states from the second system (the fifth and sixth states). From the Hessian in Figure 8 we have:

The minimal set of states for which there exists a nonzero Hessian submatrix:$\exists H_{a\mu }\neq 0$ comprises the fourth state, which we designate as the active state. The autonomous states now comprise the internal states and the first state of the second Lorenz system.The minimal set of states for which there exists a nonzero Hessian submatrix:$\exists H_{s\alpha }\neq 0$ contains the first state, which we designate as the sensory state;The particular states now comprise the autonomous states and the first state of the first Lorenz system. The blanket states therefore comprise the first states of each system;The remaining external states comprise the last pair of states of the first Lorenz system. The ensuing particular partition is shown schematically in the lower panel of Figure 10.

The key thing to note from Figure 8 is that there are profound covariances between some internal and external states, despite the fact that they are conditionally independent. Here, the second state of the first and second systems (i.e., the second and fifth states) are highly correlated and yet are twice removed, in terms of the dynamical coupling encoded in the Jacobian. We can formalise this aspect of generalised synchrony, under a particular partition, by evaluating numerically and analytically the conditional densities over the internal and external states (states two and five of the joint system) given the sensory state (i.e., state one). The lower panels of Figure 8 illustrate these densities in terms of the conditional expectations (blue lines) and 90% credible intervals (shaded areas). In this example, the densities are conditioned upon the trajectory of the sensory state based upon the trajectories from the previous figure. The broken red line in the lower panel is the external state (realised in the stochastic solution of Figure 7) that lies within the 90% credible intervals. The existence of these conditional densities has an interesting implication, which underwrites the free energy principle.

Note that there is no claim that either the original Lorenz system or coupled Lorenz system possesses a Markov blanket. The claim here is that there exists a Laplace approximation to these kinds of systems that, in virtue of the zero elements of the Hessian, feature Markov blankets.

## 5. The Free Energy Principle

The existence of a particular partition means that one can provide a stipulative definition of the conditional density over external states as being parameterised by the conditional expectation of internal states, given sensory states. We will call this a *variational density* parameterised by expected internal states, μ∈ℝ:(26)$$\begin{array}{cc} q_{\mu }(\eta ) & ≜p(\eta |\pi )=p(\eta |s)\\ \mu & ≜E[\mu |s]\\ \eta & ≜E[\eta |s] \end{array}$$

In words, for every sensory state there is a conditional density over external states and a conditional density over internal states, where external and internal states are conditionally independent. This admits the possibility of a diffeomorphic map between the sufficient statistics of the respective densities. The existence of this mapping rests upon a continuously differentiable and invertible map η=σ(μ), which is linear under a Laplace approximation. This simple form inherits from the Laplace approximation, where (treating the second and fifth states as internal and external states, respectively, i.e., $x=[s,\eta ,x_{3},a,\mu ,x_{6}]$):


(27)
$$\begin{array}{cc} q_{\mu }(\eta ) & =N(\eta ,\Sigma_{\eta |\pi })\\ \eta & =\sigma (\mu )=\mu \cdot \frac{H_{12}\;H_{45}^{2}-H_{12}\;H_{44}\;H_{55}}{H_{14}\;H_{22}\;H_{45}}=\mu \cdot \frac{\;h_{20}{}^{2}-h_{15}\;h_{21}}{h_{12}\;h_{20}}\cdot \frac{h_{5}\;}{h_{6}}\\ H_{\eta |\pi } & =H_{22}=h_{6} \end{array}$$


This analytic form of this linear synchronisation map follows from the standard expression for conditional Gaussian densities:(28)$$\begin{array}{cc} p\left(\eta |\pi \right) & =N(\eta ,\Sigma_{\eta |\pi })\\ \eta & =E[\eta ]+\Sigma_{\eta \pi }H_{\pi \pi }\left(\pi -E[\pi ]\right)\\ \Sigma_{\eta |\pi } & =\Sigma_{\eta \eta }-\Sigma_{\eta \pi }H_{\pi \pi }\Sigma_{\pi \eta } \end{array}$$

In our example, this gives:(29)$$\begin{array}{cc} \mu & =E[\mu |s]=s\cdot \frac{\;H_{14}\;H_{45}}{\;H_{44}\;H_{55}-H_{45}^{2}}=x_{1}\cdot \frac{h_{12}\;h_{20}\;}{h_{15}\;h_{21}-h_{20}{}^{2}}\\ \eta & =E[\eta |s]=-s\cdot \frac{\;H_{12}\;}{H_{22}\;}=-x_{1}\cdot \frac{h_{5}\;}{h_{6}} \end{array}$$

In short, there is a unique conditional density over external states for every expected internal state, when conditioned on the sensory state. We can leverage the existence of this conditional synchronisation map formally using a somewhat inflationary device: by definition, the Kullback–Leibler (KL) divergence between the variational density and the conditional density over external states is zero. This means the autonomous flow at the expected internal state can be expressed as a gradient flow on a free energy functional of the variational density:(30)$$\begin{array}{cc} \left[\begin{array}{c} f_{\eta }(x)\\ f_{s}(x)\\ f_{a}(x)\\ f_{\mu }(x) \end{array}\right] & =-\left[\begin{array}{c} \Gamma_{\eta }\nabla_{\eta }ℑ(x)\\ \Gamma_{s}\nabla_{s}ℑ(x)\\ \Gamma_{a}\nabla_{a}F(\pi )\\ \Gamma_{\mu }\nabla_{\mu }F(\pi ) \end{array}\right]+Q\nabla ℑ-\Lambda \\ F(\pi ) & =ℑ(\pi )+\underset{=0}{\underset{︸}{D[q_{\mu }(\eta )||p(\eta |s)]}} \end{array}$$

The free energy in question is an upper bound on the surprisal of particular states:(31)$$\begin{array}{cc} F(\pi ) & ≜\underset{Energy}{\underset{︸}{E_{q}[ℑ(\eta ,\pi )]}}-\underset{Entropy}{\underset{︸}{H[q_{\mu }(\eta )]}}\\ & =\underset{\geq 0}{\underset{︸}{D[q_{\mu }(\eta )||p(\eta |\pi )]}}+ℑ(\pi )\\ & =\underset{Accuracy}{\underset{︸}{E_{q}[ℑ(\pi |\eta )]}}+\underset{Complexity}{\underset{︸}{D[q_{\mu }(\eta )||p(\eta )]}}\geq ℑ(\pi ) \end{array}$$

This functional can be expressed in several forms; namely, an expected energy minus the entropy of the variational density, which is equivalent to the self-information associated with particular states (i.e., *surprisal*) plus the KL divergence between the variational and conditional (i.e., posterior) density. In turn, this can be decomposed into the (negative) log likelihood of particular states (i.e., *accuracy*) and the KL divergence between posterior and prior densities (i.e., *complexity*). In variational Bayesian inference [61], negative surprisal is read as a log marginal likelihood or model evidence that marginalises over external states; namely, the causes of sensory states. In this setting, negative free energy becomes an *evidence lower bound* or ELBO [30].

This is a basis of the free energy principle. Put simply, it means that the expected internal states of a particular partition—at nonequilibrium steady-state—can be cast as encoding conditional or Bayesian beliefs *about* external states. Equivalently, the flow on the internal manifold can be expressed as a gradient flow on a variational free energy that can be read as self-information. This licenses a somewhat teleological description of self-organisation as self-evidencing [34], in the sense that the surprisal or self-information that constitutes the potential is known as log model evidence or marginal likelihood in Bayesian statistics. From a physiological perspective, this is simply a statement of homeostasis [62], where effectors (i.e., active states) maintain interoceptive signals (sensory states) within characteristic physiological ranges.

### 5.1. The Generative Model and Self-Evidencing

A reading of self-information as evidence, for a particular model, begs the question: what is the model? In variational Bayes—and Bayesian inference in general—a generative model is taken to be a probabilistic specification of how causes generate consequences. In the current setting, this is just the nonequilibrium steady-state density describing the relationship between external states (causes) and particular states (consequences). In statistics, the generative model is usually decomposed into a likelihood and prior, which can be expressed in terms of their corresponding potentials:(32)$$\begin{array}{cc} ℑ(\eta ,\pi ) & =ℑ(\pi |\eta )+ℑ(\eta )\\ ℑ\left(\pi |\eta \right) & =\frac{1}{2}{\left(\pi -E[\pi |\eta ]\right)}^{T}H_{\pi |\eta }(\pi -E[\pi |\eta ])\\ ℑ(\eta ) & =\frac{1}{2}{\left(\eta -E[\eta ]\right)}^{T}H_{\eta \eta }(\eta -E[\eta ]) \end{array}$$

Here, the likelihood potential is the surprisal of particular states given external states, while the prior potential is the surprisal of external states. Under the Laplace form for the potential, these densities can be expressed in quadratic form, in terms of prediction errors and their respective precisions (i.e., Hessian matrices). Equipped with the functional form of the generative model, we can now write down the functional form of the variational free energy and its gradients. From Equation (32), we have:(33)$$\begin{array}{c} F(\pi )=\underset{Accuracy}{\underset{︸}{\frac{1}{2}{\left(\pi -E_{q}[\pi |\eta ]\right)}^{T}H_{\pi |\eta }(\pi -E_{q}[\pi |\eta ])}}+\underset{Complexity}{\underset{︸}{\frac{1}{2}{\left(\eta -E_{q}[\eta ]\right)}^{T}H_{\eta \eta }(\eta -E_{q}[\eta ])}}\\ \begin{array}{cc} \begin{array}{cc} E_{q}[\eta ] & =E[\eta |s]=\eta -s\cdot \frac{h_{5}\;}{h_{6}}\\ E_{q}[\mu |\eta ] & =E[\mu |s]=\mu =s\cdot \frac{h_{12}\;h_{20}\;}{h_{15}\;h_{21}-h_{20}{}^{2}},\\ E_{q}[a|\eta ] & =E[a|s]=a=s\cdot \frac{-h_{12}\;h_{21}\;}{h_{15}\;h_{21}-h_{20}{}^{2}} \end{array} & \;H_{\pi |\eta }=\left[\begin{array}{cccc} h_{3} & h_{12} & 0 & 0\\ h_{12} & h_{15} & h_{20} & 0\\ 0 & h_{20} & h_{21} & 0\\ 0 & 0 & 0 & h_{28} \end{array}\right] \end{array}\\ \begin{array}{cc} \nabla_{\pi }F(\pi ) & =H_{\pi |\eta }(\pi -E_{q}[\pi |\eta ])\\ & \Rightarrow \\ \left[\begin{array}{c} \nabla_{a}F\left(\pi \right)\\ \nabla_{\mu }F\left(\pi \right) \end{array}\right] & =\left[\begin{array}{ccc} h_{12} & h_{15} & h_{20}\\ 0 & h_{20} & h_{21} \end{array}\right]\left[\begin{array}{c} s-s\\ a-a\\ \mu -\mu \end{array}\right]=\left[\begin{array}{c} h_{12}\;s+h_{15}\;a+h_{20}\;\mu \\ h_{20}a+h_{21}\;\mu \end{array}\right] \end{array} \end{array}$$

Here, we have decomposed the free energy into [in]accuracy and complexity, where accuracy rests on the difference between particular states and those predicted under the variational density: i.e., prediction errors. Because the predictions are the expected particular states, given a sensory state, they can be derived directly using the standard results for Gaussians: see Equation (29) and [16]. The resulting gradients are a mixture of particular states as shown in the final equalities. The associated gradient flows can then be read as minimising the prediction errors that constitute the first term in the expression for free energy. These prediction errors correspond to the difference between any particular state and a predicted state based upon Bayesian beliefs about external states that—by construction—are the conditional expectations, given the sensory states. This means autonomous (i.e., active and internal) will look as if they are drawn towards their conditional expectations, given sensory states. When the internal state coincides with its conditional expectation (i.e., μ=μ), it encodes Bayesian beliefs about external states. This perspective on autonomous gradient flows emphasises the inference aspect of self-evidencing.

An alternative (and deflationary) perspective rests on noting that free energy gradients are also the gradients of self-information:(34)$$\begin{array}{cc} F(\pi ) & =ℑ(\pi )=\underset{Surprisal}{\underset{︸}{\frac{1}{2}{\left(\pi -E[\pi ]\right)}^{T}H_{\pi }(\pi -E[\pi ])}}\\ \pi -E[\pi ] & =\left[\begin{array}{c} s\\ a\\ \mu \\ x_{6}-\frac{h_{22}}{h_{28}} \end{array}\right],\;H_{\pi }=\left[\begin{array}{cccc} \frac{h_{3}\;h_{6}-h_{5}{}^{2}}{h_{6}} & h_{12} & 0 & 0\\ h_{12} & h_{15} & h_{20} & 0\\ 0 & h_{20} & h_{21} & 0\\ 0 & 0 & 0 & h_{28} \end{array}\right]\\ \nabla_{\pi }ℑ(\pi ) & =H_{\pi }(\pi -E[\pi ])\\ & \Rightarrow \\ \left[\begin{array}{c} \nabla_{a}F\left(\pi \right)\\ \nabla_{\mu }F\left(\pi \right) \end{array}\right] & =\left[\begin{array}{ccc} h_{12} & h_{15} & h_{20}\\ 0 & h_{20} & h_{21} \end{array}\right]\left[\begin{array}{c} s\\ a\\ \mu \end{array}\right]=\left[\begin{array}{c} h_{12}\;s+h_{15}\;a+h_{20}\;\mu \\ h_{20}a+h_{21}\;\mu \end{array}\right]\\ f_{a}\left(x\right) & =-\Gamma_{aa}\left(h_{12}\;s+h_{15}\;a+h_{20}\;\mu \right)+\ldots \\ f_{\mu }\left(x\right) & =-\Gamma_{\mu \mu }\left(h_{20}a+h_{21}\;\mu \right)+\ldots \end{array}$$

This formulation of gradient flows is simpler and shows that they are effectively minimising a different sort of prediction error; namely, the difference between particular states and the expected values at nonequilibrium steady-state. Similarly, the precision is now the precision of the marginal density over particular states, which is identical to the conditional precision above, for and only for the autonomous states. This leads to exactly the same gradient flows but a complementary interpretation, in which autonomous states are drawn towards their steady-state expectation. Heuristically, one could imagine this kind of stochastic chaos as apt to describe the motion of a moth attracted towards a flame, but being constantly thwarted by turbulent (i.e., solenoidal) air currents. Because active states influence sensory states (and possibly external states) this would look as if the particle (e.g., moth) was trying to attain its most likely state in the face of random fluctuations and solenoidal dynamics. This perspective emphasises the active part of self-evidencing, sometimes referred to as active inference [63].

### 5.2. Summary

In this section we have seen that it is fairly straightforward to construct a Laplacian system that is quadratic in both its potential function and state-dependent flow operators that evinces stochastic chaos. In the example above, the (polynomial) parameters of this functional form were chosen to emulate sparsely coupled Lorenz systems. The sparse coupling—implicit in zero entries in the Jacobian—entails a sparsity in the Hessian. This leads to conditional independencies that define a particular partition into internal, external and blanket (i.e., sensory and active) states. We have further seen that the generalised synchronisation between sparsely coupled stochastic chaotic systems induces a conditional synchronisation map that, in the example above, had a simple (i.e., linear) form. This is important because in these Laplacian systems the conditional synchronisation map is necessarily diffeomorphic. This means that for every sensory state there is a unique expected internal state and conditional density over external states. This conditional density can then be defined as being parameterised by the expected internal state (on an internal manifold) leading to a reformulation of identical synchronisation as gradient flows on a variational free energy. The resulting formulation can be read as Bayesian mechanics, in which internal states—on average—parameterise beliefs of a Bayesian sort about external states. Note that this parameterisation or elemental representation is in terms of expected internal states given sensory states. This means that to see this kind of behaviour empirically, one would have to average the internal states, given specific sensory states. Interestingly, this is almost universally the procedure adopted in the neurosciences when trying to characterise the functional anatomy or specialisation of internal states such as the neuronal states of our brains [64].

Clearly, the worked example based on sparsely coupled Lorenz systems does not mean that a conditional synchronisation map exists—or is indeed invertible—in any given system. However, the above derivations can be taken as existence of proof that such manifolds (and accompanying variational free energy formulation) can emerge from sufficiently sparse coupling.

## 6. Discussion

In conclusion, we have rehearsed a generic approach to reparameterising stochastic chaos in terms of Laplacian systems whose flow can be decomposed into a flow operator and the gradients of self-information, i.e., a potential function corresponding to the logarithm of the nonequilibrium steady-state density. This means that it is possible to describe, or at least approximate, stochastic chaos in terms of state-dependent solenoidal flow on the isocontours of a Gaussian steady-state density. This enables one to identify conditional dependencies with zero entries in the Hessian matrix—which inherit from sparse coupling, i.e., zero entries in the Jacobian matrix. In turn, this allows one to carve up state-space using particular partitions into internal and external states that are separated, statistically, by blanket states. A particular partition is necessary to talk about states that are internal to some particle or person, and which can be distinguished from external states. One way of understanding the ensuing coupling between internal and external states is in terms of generalised synchrony [37,56,57,65]. This synchronisation can be expressed as a variational principle of least action, using the notion of variational free energy.

The use of quadratic approximations to the Lorenz system in modelling communication and (generalised) synchrony is fairly established in the free energy principle literature: for example, the modelling of birdsong recognition using generalised Lotka–Volterra and Lorenz systems as generative models [66,67,68]. These models were equipped with a deep or hierarchical structure by using the states of a slow Lorenz system to modulate the parameters of a second, fast Lorenz system generating simulated sonograms. Using standard Bayesian schemes—formulated as gradient flows on variational free energy—it was possible to reproduce perceptual dynamics of the sort that the brain might use in predicting auditory streams with deep temporal structure. Subsequent work looked at coupling two synthetic birds to harness identical synchronisation as a metaphor for communication [69,70]. More recently, this work has been extended so that one system can learn from another or, indeed, many others [71]. Although these simulations may seem far removed from the treatment of stochastic chaos above, they are based on exactly the same principles; namely, interpreting stochastic flows in random dynamical systems at nonequilibrium steady-state in terms of gradient flows on variational free energy. The only difference between the songbird simulations and the simulations above was that in the songbird simulations the gradient flows were evaluated explicitly in terms of variational free energy gradients that follow from a predefined generative model (i.e., steady-state density), as opposed to deriving the steady-state density from some stochastic equations of motion, as featured in the present work.

The polynomial form of the Helmholtz decomposition in (5) may provide a generic model for observed random dynamical systems. In other words, it could form the basis of a forward or generative model that explains some empirically observed flow (estimated using the first and second moments to quantify the flow and covariance of random fluctuations over state-space, respectively). This kind of generative model is appealing because of its parameterisation in terms of the underlying nonequilibrium steady-state density. In other words, one could, in principle, try to explain empirical data in terms of a Gaussian steady-state density, which may include constraints on conditional dependencies. On one view, this is the procedure illustrated above; namely, assume a functional form for a generative model of dynamics and then optimise the parameters of this form to explain some dynamics in a maximum likelihood sense—a possible solution to the central engineering problem known as nonlinear system identification [72,73,74,75]. But how would this scale to high dimensional systems? There are some interesting issues here. First, the Lorenz system was itself designed to explain high dimensional flows [18]. By reverse engineering the construction of the Lorenz system (e.g., spectral Galerkin approximation), one can conceive of a generative model—based upon (5)—that is equipped with likelihood mappings which generate observable outcomes with a much higher dimensionality than the low dimensional space supporting stochastic chaos.

Another option for scaling this approach to high-dimensional dynamical systems, would be to *learn* the state-dependency of the flow operator and the nonequilibrium steady-state density using unsupervised learning approaches such as deep neural networks [76,77,78,79]. This would impose minimal assumptions on the functional form of the mapping between the states and the flow operator or steady-state density, while simultaneously allowing these dependencies to be fit from high-dimensional observations. This follows from the capacity for deep neural networks to represent arbitrary nonlinear relationships between observed data (here, samples of the true system’s flow) and latent variables, e.g., the elements of the flow operator and the steady-state density [76,79]. A similar approach has already become popular in the deep learning literature in the form of “neural stochastic differential equations”, which parameterise the drift and diffusion terms of a stochastic differential equation using feedforward neural networks [80]. In the same vein, one could use a separate feedforward neural network to parameterise the different components of the flow operator and the self-information. This would require (differentiable) transforms to be applied to the output layers of the network to constrain the Hessian and solenoidal flow operator to be positive definite and anti-symmetric, respectively.

It is interesting to relate flows across a Markov boundary to constructal law, which, like the free energy principle, seeks a normative account of the structure and dynamics of complex systems. In the case of constructal law, this is articulated in terms of maximising external access to the currents internal to the system [81]. Although constructal law is formulated primarily in thermodynamic terms, it should be possible to express it in information-theoretic terms by quantifying the information transfer of thermodynamic flows [82]. In this context, it would be particularly interesting to address the importance of the solenoidal flow, that underlies nonequilibria, in maximising information transfer across a Markov boundary. Perhaps one of the most remarkable conclusions from this work is that it is possible to construct systems that exhibit canonical attractors using quadratic potential functions or Gaussian steady-state densities. One practical implication of this is that it licenses the use of the (inverse) sample covariance matrix (of a sufficiently long timeseries) as a potentially sufficient description of the steady-state density. In other words, one can assume a Laplace approximation to any stochastic chaotic dynamics and use the empirical covariance matrix to identify conditional independencies from zero entries in the corresponding precision or Hessian matrix. Indeed, in principle, it should be possible to use the sample mean and covariance to specify the polynomial parameters of the potential function and then fit the observed flow at different points in state-space (by estimating the polynomial coefficients of the flow operator). This would furnish a description of the expected flow and Lyapunov exponents—to establish whether the system was chaotic or not. In conclusion, having a simple functional form for the flow of random dynamical systems may be useful for both modelling and analysing timeseries generated by real-world processes that are far from equilibrium that may or may not be chaotic.

## Figures and Tables

**Figure 1 entropy-23-01220-f001:**
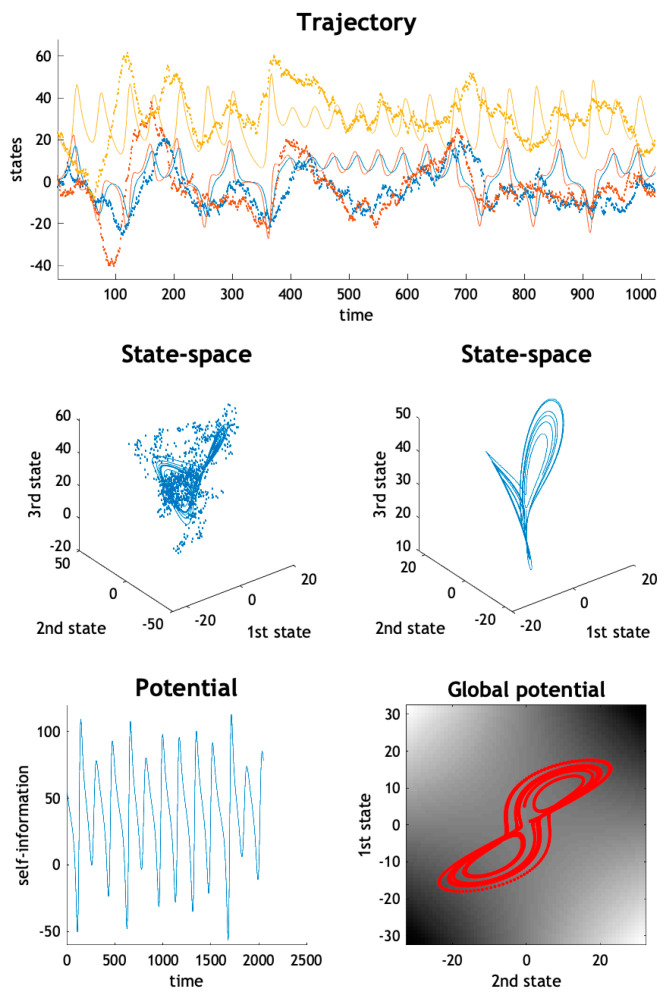
The Lorenz system and stochastic chaos. (**Upper panel**) this illustrates the solution to the Lorenz system of equations with (lines of asterisks) and without (solid lines) random fluctuations, with a variance of 16. (**Middle panels**) the left panel shows the corresponding solutions in the three-dimensional state-space, illustrating the butterfly shape of the limit set (deterministic solution: solid line) and random attractor (stochastic solution: line of asterisks). The trajectory in the right panel is the deterministic solution to the Laplacian form of the Lorenz system based upon a Helmholtz decomposition parameterised with a second-order polynomial. (**Lower left panel**) this plots the fluctuations in the potential evaluated using the Laplacian form, which expresses the self-information (i.e., potential) as an analytic (second-order polynomial) function of the states. (**Lower right panel**) this potential function (of the first two states) is shown as an image, with the trajectory superimposed.

**Figure 2 entropy-23-01220-f002:**
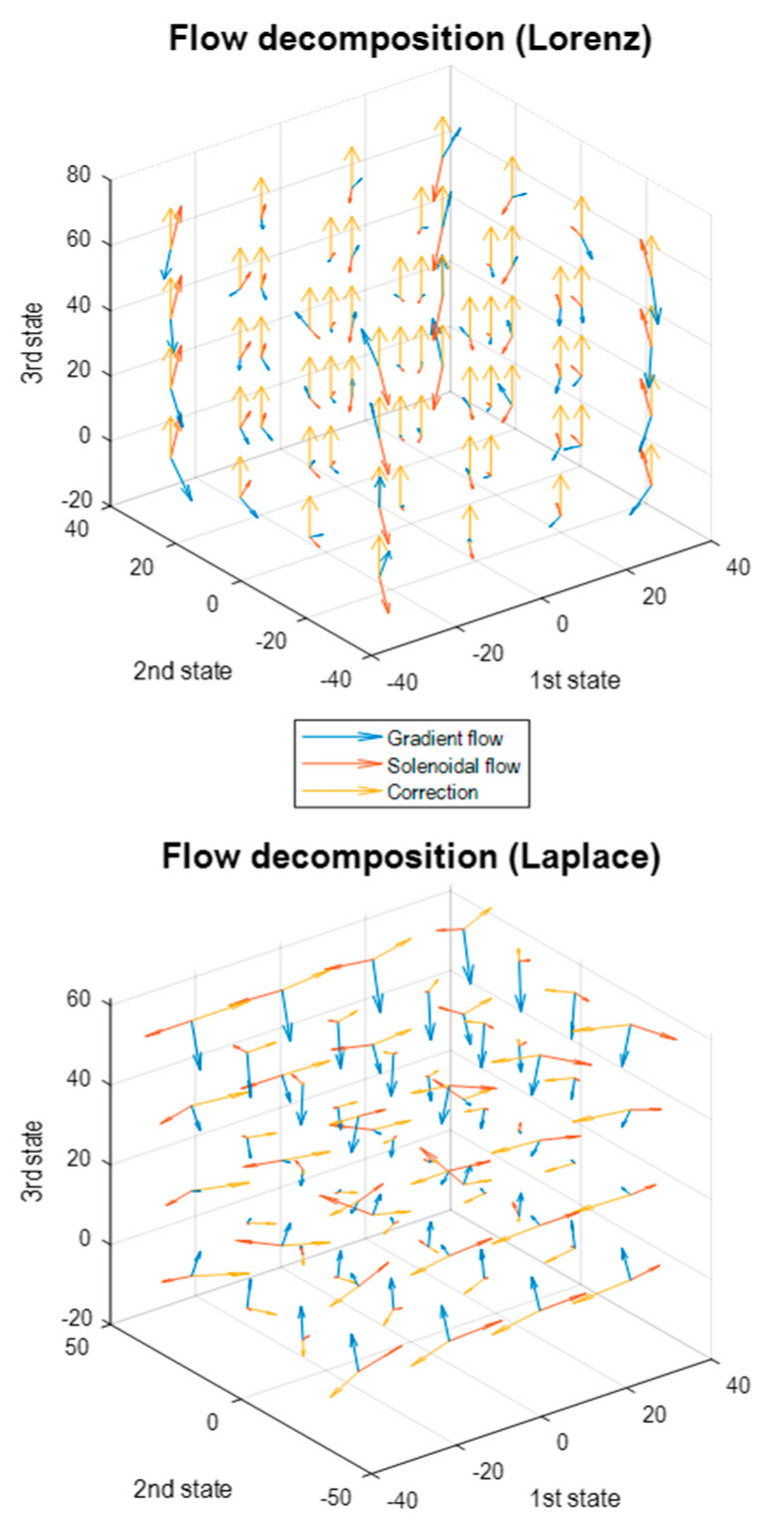
The Helmholtz decomposition of flows. (**Upper panel**) this illustrates the flow of the Lorenz system using the Helmholtz decomposition into solenoidal flow (red) gradient flow (blue) and correction flow (gold). The flow is shown as a quiver plot at equally spaced points in state-space. (**Lower panel**) this uses the same format but for a Laplacian system based upon the Lorenz system in the upper panel. The key difference here is that the dissipative part of the flow operator and Hessian are positive definite, which means the gradient flows converge to the maximum of the nonequilibrium steady-state density. This is reflected in the blue arrows that point to the centre of this state-space.

**Figure 3 entropy-23-01220-f003:**
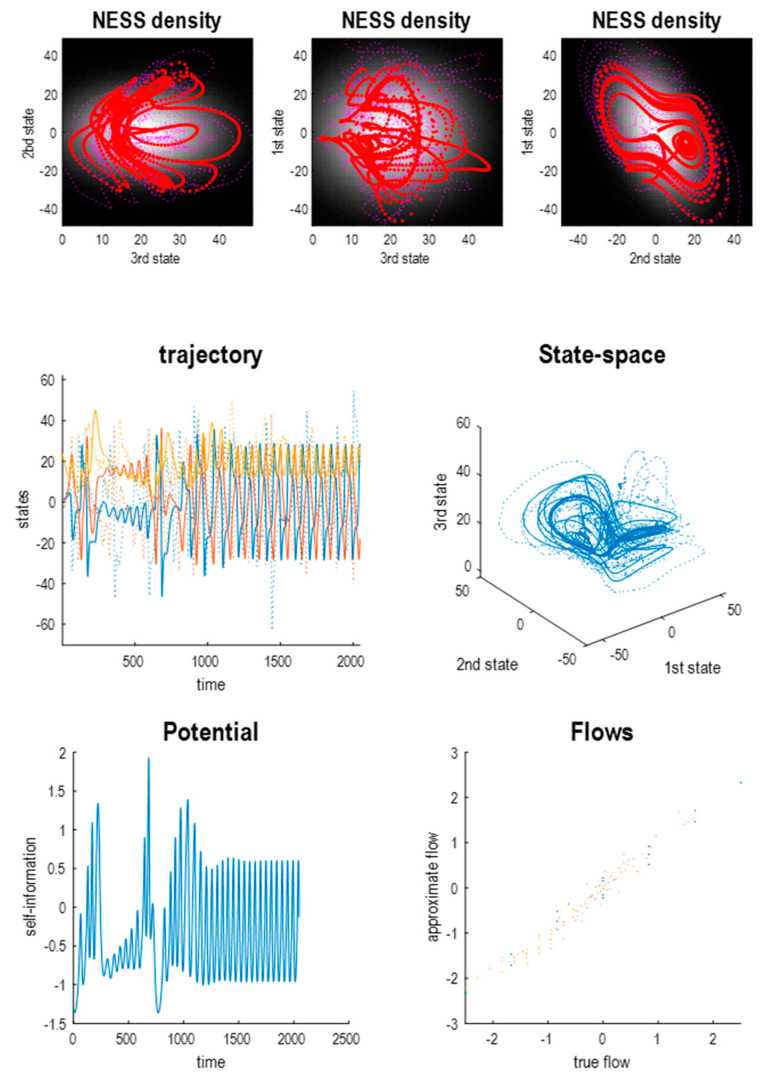
A chaotic Laplacian system. (**Upper panels**) these show the solution or trajectory of three states comprising a Laplacian approximation to a stochastic Lorenz system. The red dots mark a deterministic solution to the equations of motion, while the purple dots illustrate a stochastic solution with random fluctuations. These trajectories are superimposed on an image representation of the nonequilibrium steady-state density that—by construction in this Laplacian system—is multivariate Gaussian. (**Middle panels**) the left panel shows the deterministic (solid lines) and stochastic (dotted lines) solutions as a function of time, while the right panel plots the same trajectories in state-space. The shape of the attractor retains a butterfly-like form but is clearly different from the Lorenz attractor. (**Lower left panel**) this plots the potential or self-information as a function of time based upon the analytic form for the equations of motion and the deterministic trajectory of the previous panel. In the absence of the correction term, the gradient flow would ensure that this potential decreased over time, because solenoidal flow is divergence free (i.e., is conservative). However, there are slight fluctuations around the minimum potential induced by the correction term. (**Lower right panel**) this plots the flow of the Laplacian system (approximate flow) against the flow of the Lorenz system (true flow) evaluated at 64 equally spaced sample points. The different colours correspond to the components of flow or motion in the three dimensions. It can be seen that although there is a high correlation between the flows of the Laplacian and Lorenz systems, they are not identical.

**Figure 4 entropy-23-01220-f004:**
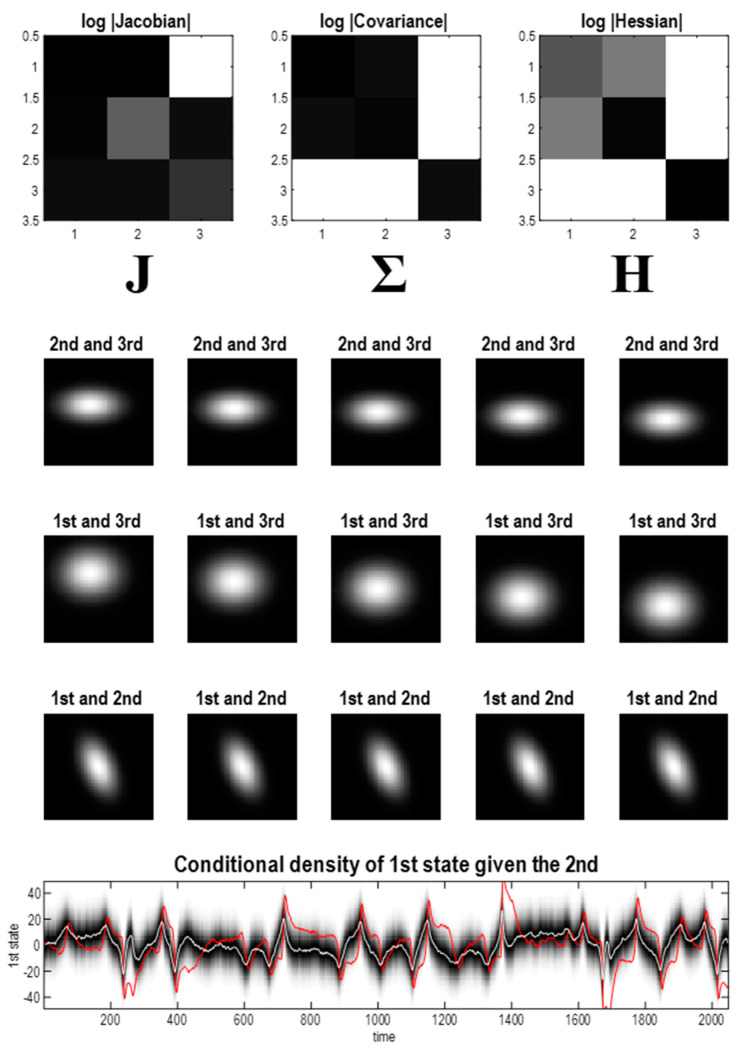
Nonequilibrium steady-state density. (**Upper panel**) these images report the constraints on coupling entailed by the Jacobian (left) and manifest in terms of the Hessian (right). The inverse of the Hessian matrix can be read (under the Laplace assumption) as the covariance matrix of the three states. In this example, the third state is independent of the first pair, where this independence rests on the directed coupling from the third to the first state. The matrices correspond to the log of the absolute values of the matrix elements—to disclose their sparsity structure. (**Middle panel**) these show slices through the ensuing steady-state density over two states, at increasing values of the remaining state. They illustrate the fact that the only correlation in play is between the first and second states. (**Lower panel**) this correlation is illustrated in terms of the conditional density over the first state, given the second. The shaded areas correspond to the probability density and the white line is the conditional expectation. The red line is the realised trajectory of the first state that is largely confined to the 90% credible intervals. This characterisation uses the trajectory from the stochastic solution shown in Figure 3.

**Figure 5 entropy-23-01220-f005:**
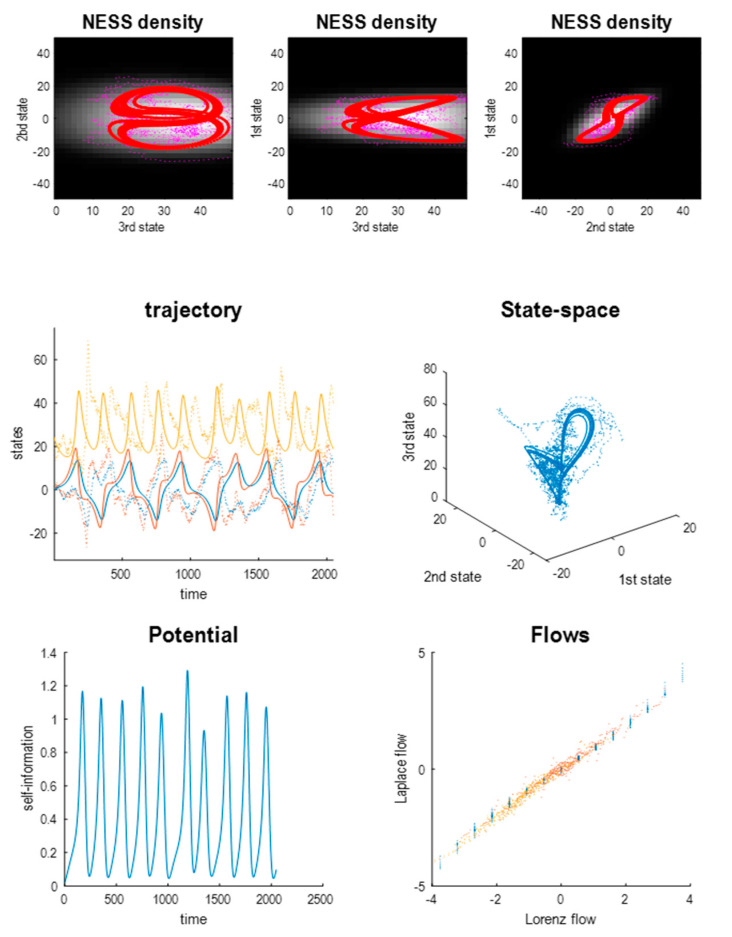
A high-order approximation. This figure uses the same format as Figure 3 to illustrate the dynamics of a three-dimensional system that is indistinguishable from a Lorentz system. However, in this instance, the equations of motion can be decomposed into a solenoidal and gradient flow in which the dissipative part of the flow operator and Hessian are positive definite. In other words, this system is apt to describe stochastic chaos driven by random fluctuations to a proper nonequilibrium steady-state density. In this example, the solenoidal flow was parameterised up to second-order and the potential up to fourth-order, with constraints to ensure the Hessian was positive definite everywhere. The high order terms in the Hessian mean that the steady-state density in the upper panels is no longer Gaussian (although univariate and bivariate conditional densities remain Gaussian).

**Figure 6 entropy-23-01220-f006:**
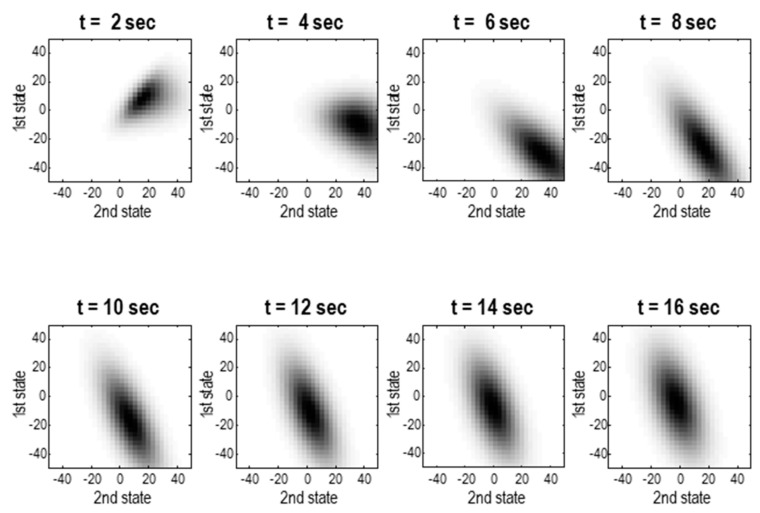
Density dynamics. These images show snapshots of a time-dependent probability density for the chaotic Laplacian system in Figure 3. They report the marginal density over the first two states, averaged over successive epochs of two seconds (assuming an integration time of 1/64 s). The system was prepared in an initial state with a relatively precise density, centred around [4, 4, 8]. This density converges to the steady-state density (see Figure 3) after about 16 s; however, it takes a rather circuitous route from this particular set of initial states. Note that the average density over short periods of time can be highly non-Gaussian, even though the density at any point in time is, by construction, Gaussian.

**Figure 7 entropy-23-01220-f007:**
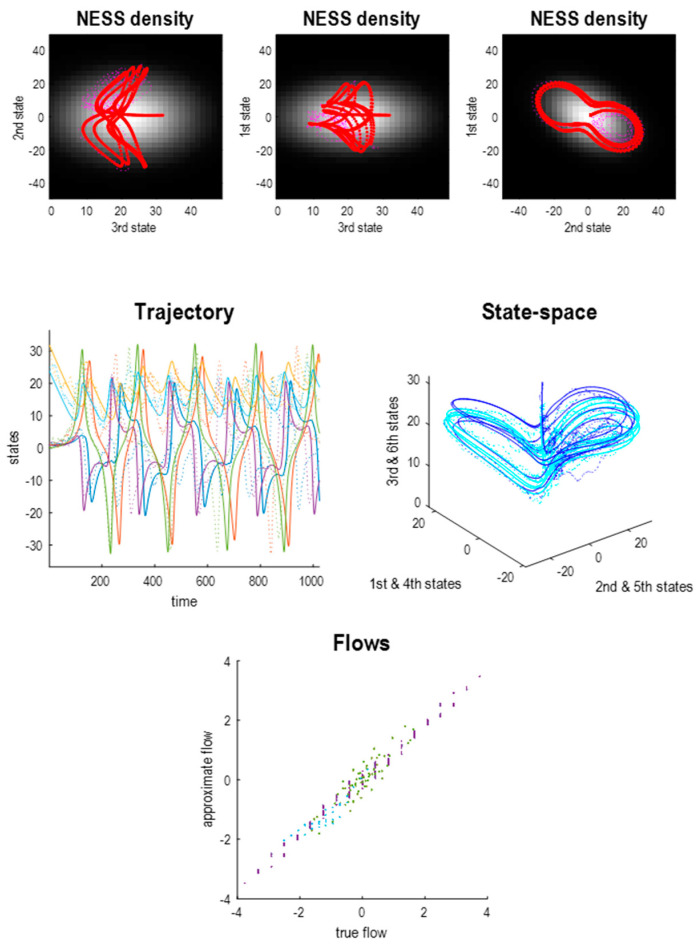
Generalised synchrony in a Laplace system. (**Upper panels**) This figure uses the same format as Figure 3 but, in this instance, reporting the Laplacian approximation to coupled Lorenz systems evincing generalised synchrony. (**Middle panels**) The deterministic (solid lines) and stochastic (dotted lines) solutions of this six-dimensional system are shown in a three-dimensional state-space by plotting the three states of the coupled systems on the same axes (in blue and cyan, respectively). This illustrates the degree of synchronisation, which is particularly marked for the deterministic solutions (corresponding to identical synchronisation). (**Lower panel**) As in Figure 3, the flow of the Lorenz (true) and Laplace (approximate) systems are not identical but highly correlated.

**Figure 8 entropy-23-01220-f008:**
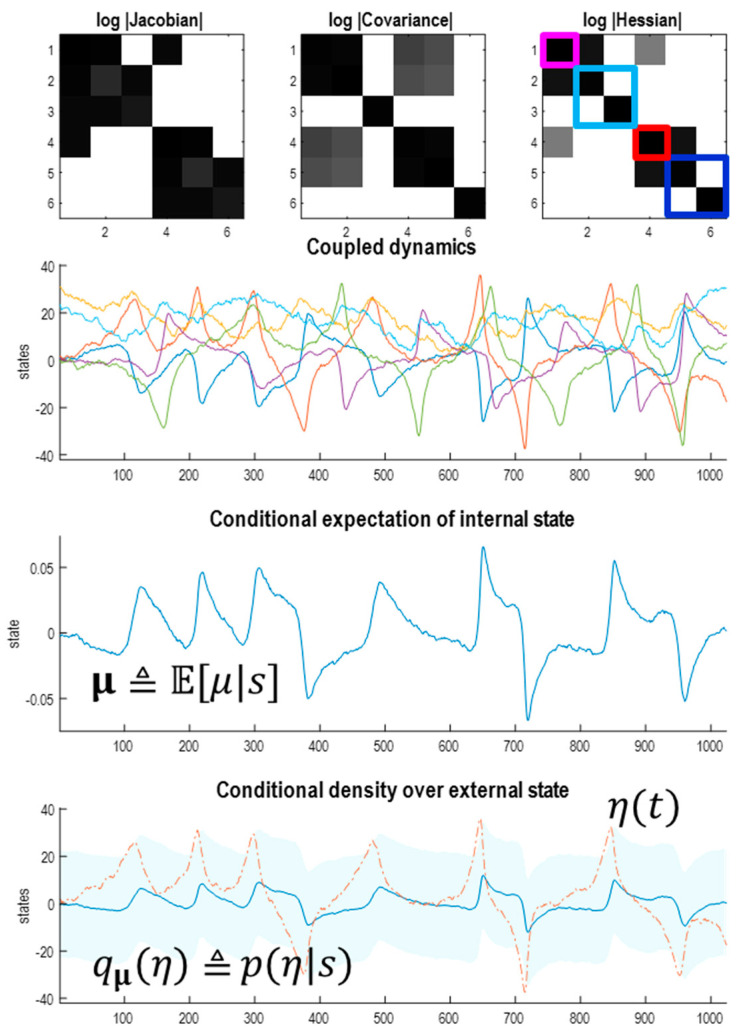
Variational inference. (**Upper panel**) these images use the same format as Figure 4 to illustrate the sparsity of the coupling (in the Jacobian: left) and ensuing conditional independencies (in the Hessian: right). This sparsity structure now supports a particular partition into internal (states five and six), active (fourth state), sensory (first state) and external (second and third) states. This partition is illustrated with boxes over the Hessian: blue—internal states, red—active states, magenta—sensory states and cyan—external states. The remarkable thing here is that despite their conditional independence there are correlations between internal and external states and here, between the second and fifth states. (**Second panel**) this plots a stochastic solution of all six states as a function of time. (**Lower panels**) the correlations between the fourth (internal) and second (external) states imply one can be predicted from the other. This is illustrated by plotting the conditional expectation of the internal state, given the sensory (first) state, in the third panel, and the associated conditional density over the external state in the fourth panel. The conditional density is shown in terms of the conditional expectation (blue line) and 90% credible intervals (shaded area). The red line corresponds to the realised trajectory of the external state that lies largely within the 90% credible intervals.

**Figure 9 entropy-23-01220-f009:**
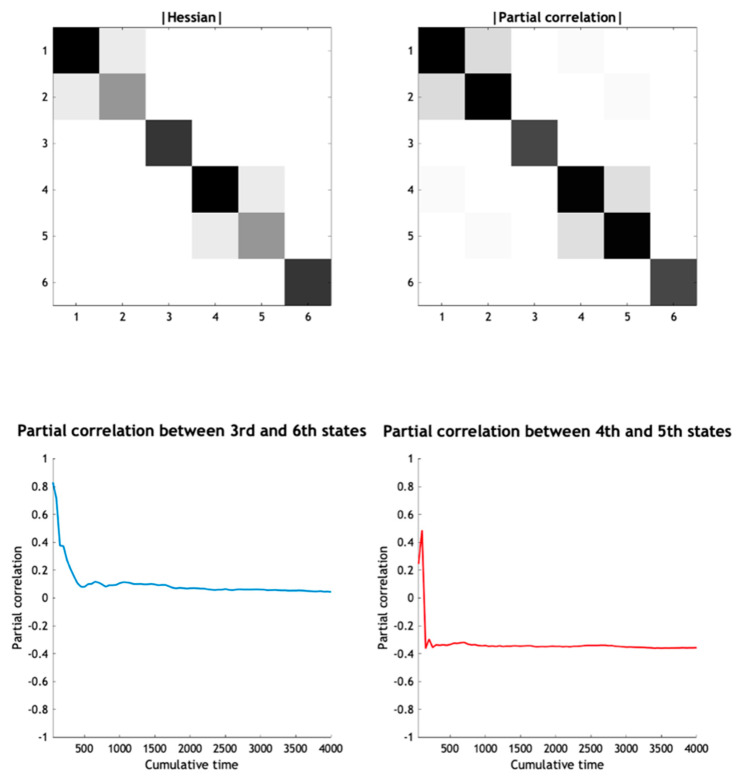
A numerical analysis of conditional independence. (**Upper panel**) The partial correlation $P_{XY\cdot Z}$ (right) between each pair of states, regressing out the effect of the remaining states, approximates the Hessian of the (Laplace-approximated) coupled Lorenz system (left). In this figure, the Hessian and the partial correlation are displayed in terms of their norms (i.e., the elements squared). The partial correlation matrix was based on 128 stochastic solutions, each lasting 500 s. The upper right panel shows the average partial correlation matrix based on the entire timeseries and averaged over realisations. (**Lower panel**) partial correlations are shown for two pairs of states—the 3rd and 6th states (the second dimensions of the respective “internal states” of the first and second Lorenz systems, left side) and the 4th and 5th states (the “sensory state” and first dimension of the “internal state” of the second Lorenz system, right side). The *x*-axis denotes the increasing length of the timeseries used to evaluate the partial correlations. Note that the 4th and 5th states are not conditionally independent, explaining why the average partial correlation converges to a value around −0.33, whereas the 3rd and 6th states are conditionally independent, given the other states, meaning that the partial correlation converges to 0.

**Figure 10 entropy-23-01220-f010:**
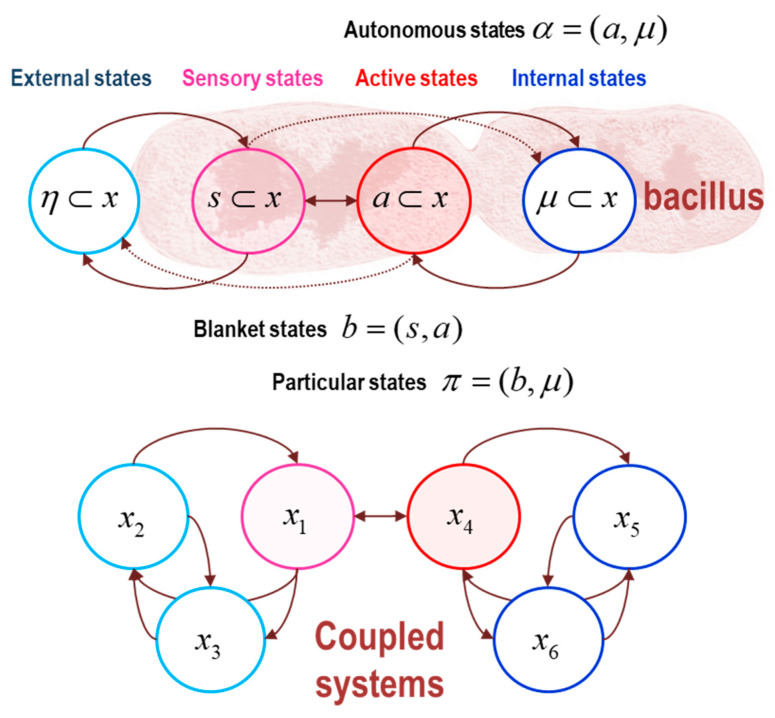
Markov blankets. These influence diagrams illustrate a particular partition of states into internal states (blue) and hidden or external states (cyan) that are separated by a Markov blanket comprising sensory (magenta) and active states (red). The upper panel shows this partition as it would be applied to a single-cell organism, where internal states are associated with the intracellular states, the sensory states become the surface states or cell membrane overlying active states (e.g., the actin filaments of the cytoskeleton). The dotted lines indicate directed influences from external (respectively internal) to active (respectively sensory) states. Particular states constitute a particle; namely, autonomous and sensory states—or blanket and internal states. The lower panel illustrates how this partition applies to the six states of the coupled system considered in the main text.

## Data Availability

The code used to run the simulations and create the figures for this paper are freely available in the DEM toolbox of the MATLAB package SPM12, which can be downloaded here: https://www.fil.ion.ucl.ac.uk/spm/software/spm12/. The relevant script within the DEM toolbox is called FEP_lorenz_surprise.m.

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
