# Peer review of "Stochastic Chaos and Markov Blankets"

_entropy, 2021, doi:10.3390/e23091220_

Round 1

Reviewer 1 Report

The paper is very interesting, but some improvement can be suggested.

In the paper, the role of fluxes is introduced. I suggest to introduce some comments on Constructal law from Bejan, Lorente, and other authors papers. For example in relation to synchronisation there exist the problem of definition of time. In Scientific Reports a definition of time related to Constructal law has been developed, and in Materials its extension to non-equilibrium themperature was introduced by Grisolia et al. Moreover, in Entropy journal some comments on this topic has been developed by Kuzemsky et al. in order to develop just the definition of time in relation to entropy production, exergy, and Gibbs free energy.

Moreover, on Atti dell'Accademia Peloritana dei Pericolanti some papers on the Algebraic topology in thermodynamics was developed in relation to the topic of the paper.

Last, always Grisolia et al. developed the use of fluxes in biosystems in Applied Sciences and Entropy journals, but also experimental results have been published on Roysal Sociaty and J. Theor. Biol.

Stochastic paths was also been discussed on Physica A, in relation to Janes theory and Prigogine approach to chaos and non-equilibrium thermodynamics.

Last, I suggest also to consider some results from Beretta and Sciubba.

I suggest to improve the paper by considering also these results.

After some comments on these other results, this paper could certainly be accepted.

Reviewer 2 Report

Previous work in the literature on the free energy principle (FEP) has been built on an assumption that complex systems of a certain form have 'Markov blankets' in a specific sense. However, it is not obvious that this is true - it does not follow straightforwardly - and a few recent papers have called the claim into question. The current paper tries to address this by showing that a specific system does indeed exhibit such a Markov blanket, and that the Markov blanket emerges from the structure of the system in the same way as claimed (for example) in reference 27.

As such the paper is timely and potentially important, since this aspect of the theory's foundations does need to be clarified, and a good non-trivial and non-linear worked example would shed a lot of light on it.

Unfortunately, for a technical reason, I'm skeptical that the paper has achieved what it sets out to achieve, and my recommendation is that the paper be revised to clear up the issue. In this review I first explain what this issue is and suggest one possible way to resolve it, and then I move on to some more specific issues that I think should also be addressed.

--

The problem is that, if I have understood the overall argument correctly, all of the main results are based on an approximation (the Laplace approximation) that inherently throws away all higher-order correlations. Therefore the results are only valid to the extent that the Laplace approximation is a sufficiently good approximation to the system under consideration. This is a particularly tricky issue in the context of this paper, because the Laplace approximation assumes the stationary state is a Gaussian, while the true stationary state of the Lorenz system (even with added noise) has its characteristic butterfly shape, which differs quite substantially from a Gaussian. For the results to be convincing one would have to show that in spite of this the conditional independence relationships one derives from the Gaussian can be expected to hold in the true stationary state.

If I have understood correctly, this issue applies not just to the examples but also to all of the more general results in the first part of section called "Particular partitions, boundaries and blankets", starting on line 581. These include equation 23 and the more informal statements such as "any two states are conditionally independent if at least one state does not influence the other" (line 600), or "...any sparse coupling implies a nonequilibrium steady-state with conditional independencies" (line 610). Although these are stated this way without qualification, my understanding is that the Laplace approximation is used to derive them, and hence they have strictly speaking only been shown in systems for which the Laplace approximation is exact. This seems unlikely to be the case for most nonlinear systems of interest, so it seems rather important to demonstrate that they still hold (perhaps in some suitable approximate sense) in systems for which the Laplace approximation is only an approximation.

The paper doesn't offer a specific argument to justify the approximation (aside from it making the equations simpler, which is surely true), and by leaving that out the paper leaves open the question of whether its results are purely an artefact of the approximation used to derive them. In the specific case of the Lorenz attractor (or two coupled Laplace attractors), these questions about the validity of the approximation are fairly concrete, and it seems like it should be possible to check them quite directly, as I explain below.

Although there might be other good ways to address this issue, I would like to suggest the following as one possible way to do it. First, perform a large number of Monte Carlo simulations of the Lorenz system (the original, not its Laplace approximation) with added noise, to produce a set of independently distributed samples from the stationary state. Then bin these to produce a histogram, approximating the joint distribution between the three variables. Then check directly whether the mutual information between the first and third variables is indeed close to zero. (Note that it is the mutual information that needs to be evaluated here and not just the covariance, since conditional independence refers to all correlations, not just linear ones.) If the independence relationship does hold it should approach zero as the number of sample points increases. If it instead approaches some small finite value then that might suggest that the conditional independence property holds approximately but not exactly. How good the approximation is might depend on the amount of noise added. A similar experiment could be performed for the coupled Lorenz system, measuring the appropriate conditional mutual information to check whether the claimed Markov blanket really does exist or not.

There might be other ways to address this issue, for example by quantifying how good the approximation is mathematically, or showing that it becomes exact in some limit (for example, when the amount of noise is small, or when the amount of noise is large). It does seem to me that checking the results against Monte Carlo simulations would be the most convincing way to demonstrate their validity, though, and I do strongly encourage the authors to explore something along these lines, even if the validity of the approximation can also be studied mathematically.

Of course another possible approach, which I think parts of the paper take, is to say that the example under consideration is not the Lorenz system itself but only its Laplace approximation. However, this would substantially weaken the more general results of the paper, as mentioned above. Those more general claims would need to be heavily qualified to make clear that they only hold under a very strong and somewhat unnatural condition on the dynamics (exactness of the Laplace approximation) and might not hold in general.

--

A closely related issue is that what seems to be a crucial technical point never really gets explained. I refer here to the first "<=>" in equation 14, which appears in another form as the last "<=>" in equation 22. At first I didn't think this could possibly be true, but I later realised (I think) that it is intended as a statement about the Laplace approximation, rather than about stochastic differential equations in general. But even as a property of the Laplace approximation it doesn't seem straightforward, and it's the sort of claim for which I would like to see a clear statement (including the assumptions needed for it to hold), together with a proof or a reference to a proof. This claim seems to be the basis of all of the main results of the paper, so it seems quite important to state and prove it clearly.

Here is my reasoning for thinking it's a claim about the Laplace approximation, rather than a general claim. In general, I(x) = -log p(x) is a more or less arbitrary function of x, and hence a condition on its second derivatives can't possibly be equivalent to a condition on the whole function, which is what the stated conditional independence condition amounts to. However, if we take the Laplace approximation then the approximate version of I(x) is assumed to be quadratic, so it becomes plausible that a condition on its second derivatives could be equivalent to a condition on the whole approximated distribution. But even in that case the statement doesn't seem straightfoward and I would like to see a proof.

--

The remainder of this review consists of comments on specific paragraphs that the authors might want to consider for improvement, as well as minor errors.

lines 62-75. The references in this paragraph are a mixture of papers about two different topics, namely stochastic differential equations on the one hand and the Lorenz system on the other, the latter being traditionally defined as a system of ordinary ODEs without a noise term. This mixing of topics makes it difficult for the reader to undetstand which references are relevant to which claims. There is a claim that the Lorenz system has a pullback attractor, but the citations in that sentence are about stochastic systems and don't mention the Lorenz system. Further down a claim is made about nonequilibrium and the loss of detailed balance, but this is backed up by papers about deterministic chaos that don't mention detailed balance. It would be helpful to restructure the paragraph to make it clear which claims are backed up by which referenes, and to make clear the relationship between the references on deterministic chaos and the references on stochastic dynamics.

line 72: the reference to "Arnold and SpringerLink" is obviously a typo and seems to be a duplicate of the Arnold reference on the same line.

line 100: "the existence of this manifold..." - no manifold was mentioned before this point. (Unless a "synchronisation map" is a manifold? In which case it would be helpful to say so.)

line 123: "[stochastic systems with pullback attractors] can be described with stochastic differential equations" - is this true? I was more under the impression that systems with pullback attractors were more general than stochastic differential equations.

line 138: what does "(usually strange)" mean here, precisely? (Does it mean "usually in practice", or is it a mathematical claim?)

line 141: "If this flow shows exponential divergence of trajectories, we can impute stochastic chaos" - while optional, it would be nice to have a brief description of how stochastic chaos differs from deterministic chaos.

I would split eq. 2 into two equations, or at least mention that it's really two equations in one - the left-hand one is the equations of motion and the right-hand one is derived from them. (This might not be obvious to an inexperienced reader.)

The dotted lines in the top and middle parts of fig. 1 are hard to make out. They make it hard to see whether the stochastic solution has extra small-scale wiggles that are lacking in the deterministic solution. I wonder whether colour or some other method could be used to distinguish them instead.

Line 178 says to see the appendix for an introduction to the Helmholtz decomposition, but the appendix begins by referencing equation 8, which is on line 241. The appendix does contain a sort of heuristic derivation of the more general equation 3 (on line 179), but no real explanation is given, so it is not helpful as an introduction. (It begins by defining the term Lambda, but only in terms of Q and Gamma, whose meaning has not been stated at this point.) It would not hurt to repeat the statement of lemma B.1 from the cited reference (using the notation of the current paper, and referring to the reference for the proof), since this states quite clearly what the point is.

Lines 190-195: this sounds quite odd to me. If we take the deterministic limit by just setting the noise terms to 0, we are left with a general set of ordinary differential equations. But in that case there's no reason why these equations should be Hamiltonian, i.e. that the flow should be divergence-free. The paragraph states that "as the amplitude of random fluctuations decreases, the rotational flow comes to dominate". However, it is not obvious that this is the case, since the magnitiude of the other terms will also depend on the amplitude of the fluctuations. Could the claims here be backed up with proofs or more detailed references? (Or possibly removed, since I think they aren't crucial to the rest of the paper.)

I didn't clearly understand the point about state-dependent changes in amplitude on lines 212-213, so this could be expanded upon.

Equation 5 and equation 8: I am unsure about this, but shouldn't the polynomials for I(x) have constant terms, to make sure the distribution is normalised?

In equation 8, what precisely is meant by "the Lorenz system" here? We have only been given a deterministic equation (equation 2), but here we are also talking about fluctuations, so I guess this is referring to some stochastic version of it. It would be helpful to write down the equations for this explicitly.

But then, around lines 260-268 it seems to say that the analysis fails because we're considering a system without fluctuations. So I was a bit confused by this part of the paper. Are fluctuations being considered at this point, and if they aren't, how is the generalised Helmholtz decomposition derived?

lines 245-246, "Figure 1 (right middle 245 panel) shows a solution to the above Laplace version, which is indistinguishable from the Lorenz system" - is this indistinguishability a quantitative claim, or does it just mean it looks visually similar? If the latter, this is hard to judge without a corresponding plot of the true Lorenz system from the same angle.

equation 13: u and v seem to be used here for the first time and it's not at all clear what they refer to. Are they new names for x1 and x2 in eq. 2, or are they generic indices, or something else? (I am assuming from eq. 14 that they are meant as generic indices, but here they're introduced via the specific example of x1 and x3 in the Lorenz system, so it's a bit confusing.)

Line 380-381: "Zero Hessian elements ... imply conditional independence" contradicts with "In other words, if two states are conditionally independent ... then their corresponding entries in the Hessian are zero". This second statement says that conditional independence implies zero Hessian entries, while the first says that zero Hessian entries imply conditional independence. Equation 14 seems to say that both are true, but this logical error in the text should be corrected - the two statements aren't the same.

line 387, should "conditionally independent" be "independent" here?

on line 631 there is a reference to "the literature" with no citations given.

On line 792 the text "Error! Reference source not found" appears.

In the appendix, the paragraph starting at line 1000 ("In this appendix we use the notation...") would make more sense before the notation is used rather than at the end.

Round 2

Reviewer 2 Report

Thank you to the authors for their fast and helpful response. The clarification about equation 14 was very helpful, and many of my other concerns have been resolved, as I explain in detail below.

However, in reexamining the paper I have become convinced that one of its substantial claims is not correct. This is the claim below eq. 22 that "any two states are conditionally independent [conditioned on all other variables] *if at least one state does not influence the other* – i.e. nonreciprocal coupling" (line 616, original emphasis), where "at least one state does not influence the other" means either J_uv = 0 or J_vu = 0.

I didn't think this claim could be true in general, and I originally thought the issue must have to do with the Laplace approximation. After the authors' clarification I understand that this isn't the case - the Laplace approximation isn't used in deriving the claim, and the claim is supposed to hold in general, for any stochastic differential equations.

However, it still didn't seem like it could be true. Non-reciprocal interaction generally does give rise to correlations in other types of stochastic system such as discrete-time Markov processes and jump processes, and it would be surprising if stochastic differential equations would behave differently in that respect. So I tried checking it on the simplest example I could come up with, and found that indeed it doesn't hold even in that case. I believe that the reason for this is that the proof given doesn't work, which means the claimed result isn't true.

Here is the simple example where it fails:

dx/dt = -ax + v(t)
dy/dt = -by + x + w(t)

Here v(t) and w(t) are Wiener processes, and a>0, b>0. This is a case of non-reciprocal interaction because y does not influence x, that is, J_xy=0. The claim in the paper implies that for this system x and y should be independent in the stationary state of this process.

At the end of this report I have shared some Python code that numerically simulates a number of independent runs of this system for long enough to reach stationary state, then plots them as a scatter plot. (See image below.) It's clear from the plot that x and y are not independent but strongly correlated. Numerically the off-diagonal terms of the covariance matrix are about as big as the diagonal ones. (In the code and below I take a=1/10 and b=1.)

I discussed this example system with a colleague, who was able to find the stationary state analytically with Mathematica. This yields a covariance matrix of [[5, 50/11], [50/11, 111/22]], which is consistent with the sample covariance matrices that my code produces. The Hessian H can be computed from the inverse of this, which also has non-zero off-diagonal entries. The Q matrix works out to be [[0, 10/11], [-10/11, 0]], so it has non-zero off-diagonal elements as well. The argument in the paper says that both Q and H should have zero off-diagonal elements for this system, so something must have gone wrong.

I think the reason is that the argument for the result is flawed. The argument proceeds by taking an equation that can be split into multiple terms (say, for example, a+b+c+d = 0) in which we know some terms are nonnegative, say a >= 0 and b >= 0. The argument tries to conclude from this alone that a=b=0, on the grounds that we can ignore the edge case in which a+b is exactly balanced by c+d. But I don't think there is any reason why this can be treated as an edge case and ignored. The case where a=b=0 seems like just as much of an edge case, since c and d still have to exactly balance each other.  More practically, in the context of stochastic differential equations, the case where the positive terms are balanced by the other terms occurs already in the simplest possible case, which demonstrates that it can't be ignored in practice. I think that rather than being an edge case that this is the most generic case, with the case where those terms are zero being the rare case that would require special conditions in order to hold.

I think this comment might be helpful to understand why this happens: in the new footnote (footnote 2 on page 21), the authors write "Heuristically, knowing the motion of something tells you nothing about where it is." But knowing the value of a variable ε time units ago tells us its current value to within a term of order ε^2, so  knowing a variable's recent past motion actually tells us a lot about where it is. In the example above, the variable y tends to increase when the variable x is high, and decrease when it's low. Because of this, if x was high in the recent past then x will tend to be high now (because x's current position is correlated with its past position) and also y will tend to be high (because of the past influence of x). This is what leads the current values of x and y to be correlated. They share a common cause, namely the past motion of x.

The failure of this result does seem to be a significant issue for the manuscript. It seems to me unlikely that the result can be easily fixed (although I'm happy to be proven wrong), and it does affect quite a few of the other results in the paper. On the other hand I don't think it affects all of the paper, and it could be that it can be removed while keeping results that aren't derived from it.

This concludes the major technical point. The points below are comments on the rest of the manuscript that should be straightforward to take account of.

--

The clarification about the issue in equation 14 is very helpful. In the review response document the authors add in a quantifier ($\forall x$) that was previosly missing. This makes a big difference! Without it I thought you were saying that the conditional indpendence relationship would hold if the Hessian had a zero entry for some particular value of x, and I couldn't understand how that could be true. With the quantifier it makes sense.

However, the quantifier is still missing in the manuscript, and would be helpful to add it there as well, to prevent others from being confused by the same point. I guess you could write "almost everywhere" or similar, instead of $\forall x$.

Even with the quantifier there are quite a few steps in going from the first line of equation 14 to the second, and it took me quite some work to see why it followed. I think it would be much better to spell this out in detail, since it still seems to be a core part of the paper and it's not obvious at all. At a minimum I would say something along the lines of the following, which is far from a complete proof but at least gives some hints to the reader about how to proceed:

"This follows from the fact that for any differentiable $h(x,y)$ we have $\left(\frac{\partial h(x,y)}{\partial x\partial y} = 0 \quad\forall x,y\right)\,\Longleftrightarrow\, h(x,y) = f(x) + g(y)$ for some functions $f, g$. In the case of $h(x_u,x_v) = \mathfrak{I}(x_u,x_v,b) = -\log p(x_u,x_v,b)$ it follows that $p(x_u,x_v,b)$ must factor into a function of $x_h$ times a function of $x_v$ for each value of $b$."

You might have a different proof, since you mentioned that the condition on the derivative only needs to hold almost everywhere. If so that's even better, but I would say the proof needs to be included in the paper (or clearly referenced) and not skipped over.

--

I also thank the authors for adding the clarification that no claim is being made about the Lorenz system or coupled Lorenz system. (Rather, the claims are only about the system that arises from taking the Laplace approximation, which is a different system and might not have the same conditional independence relationships as the original.) This resolves a lot of the issues in my previous review.

However, it would be useful to also clarify this earlier in the paper, because the introduction doesn't currently give the impression that this is how the paper will proceed. In particular, the introduction (lines 84-85) states:

"For the specific example at hand, a quadratic expansion is sufficient to approximate the flow, which means the steady-state density reduces to a Gaussian form." 

My current understanding is that the paper doesn't try to show that the approximation can be used to draw conclusions about the original system regarding Markov blankets, so I would suggest removing the claim that the approximation is sufficient. (I say this especially since, at least in the low-noise regime, we know that the steady state density of the true system isn't approximately Gaussian.) I would suggest also to add a statement in the introduction at around this point along the lines of what has been added on line 739, to make it clear that the paper doesn't try to evaluate the suitability of the approximation for drawing conclusions about the original, non-approximated systems.

--

I found it surprising that these papers aren't cited in the manuscript, as they provide background for why the relationship between sparsity constraints and Markov blankets is an important question.

Biehl, M.; Pollock, F.; Kanai, R. A Technical Critique of Some Parts of the Free Energy Principle. Entropy 2021, 23, 293.

Friston, K.; Da Costa, L.; Parr, T. Some Interesting Observations on the Free Energy Principle. Entropy 2021, 23(8), 1076.

--

Attached are the Python code and scatter plot.

Here is the Python code mentioned above:

# stationary_correlation.py # This is python code for simulating the stochastic differential equations # dx/dt = -ax + v(t) # dy/dt = -by + x + w(t) # The code simulates 100 independent instances of this system for 1000 # time units using the Euler method with dt=0.01. It calculates the # the covariance between x and y and plots the samples of x and y as a # scatter plot. It uses a=0.1 and b=1, which happens to make all the # terms in the covariance matrix about the same size. # Incresing the integration time and decreasing the time step # by a factor of 10 don't seem to make a noticeable difference, so I # believe this is close to the true stationary state. # We should expect x and y to be uncorrelated if one of the claims in the # paper is true, but instead the correlation is very pronounced. # This should run on any Python 3 installation with numpy and matplotlib # installed. Just save it in a file called stationary_correlation.py and # run # python3 stationary_correlation.py # replacing "python3" with whatever command you use to run Python 3 # (often just "python") # -- parameter settings -- # number of points to sample. An independent system is simulated # in parallel for each point, so this increases run time linearly nsystems = 100 # time step - decrease this to get a better approximation to the true # stochastic dynamics. This will increase the number of simulation steps, # so run time is proportional to 1/dt. dt = 0.01 # run time - the amount of time to simulate each system before plotting # the sample. This increases run time linearly. T = 1000 # parameters of the system we're simulating a = 0.1 b = 1 # -- end of parameter settings -- import matplotlib.pyplot as plt import numpy as np runsteps = int(T/dt) # initialise each system at x=0, y=0 x = np.zeros(nsystems) y = np.zeros(nsystems) sqrtdt = np.sqrt(dt) for t in range(runsteps): x, y = ( x + dt*(-a*x) + sqrtdt*np.random.standard_normal(nsystems), y + dt*(x - b*y) + sqrtdt*np.random.standard_normal(nsystems) ) print("sample covariance matrix:") print(np.cov(x,y)) print("inverse") print(np.linalg.inv(np.cov(x,y))) fig, ax = plt.subplots() ax.set(xlabel='x', ylabel='y', title='Samples from the stationary state of a linear stochastic process') ax.scatter(x,y) plt.show()

Round 3

Reviewer 2 Report

In my previous report I identified a flaw in the paper, which the authors have addressed by stating the result as a conjecture instead. While a conjecture isn't quite as nice as a proven result, I do still think the paper overall is interesting, and it makes a valuable contribution through the worked example it provides.

I won't insist on it, but as per my previous comment I think it would help the reader to include more details on why the first <=> follows in equation 14. (This is a separate issue from the missing quantifier, which has been added.) I agree that it does follow, it's just that it isn't obvious and it would be helpful to either spell it out or give a reference to more details.

Aside from that, the following are relatively minor comments and are only about changes since the last revision.

--

General comment: I note in passing that it's easy to add nonlinear terms to the code I provided previously and it seems the same thing can happen in nonlinear systems as well. So for the conjecture to hold it must be important that the system is high-dimensional as well as nonlinear. If it's possible to say anything about why the dimensionality might make a difference in this way, that would be helpful.

--

Clarity issue: "One can conjecture that the number of such solutions is vanishingly small for sufficiently high dimensional and nonlinear systems. If we ignore these (edge) cases, ..." - this sounds like it's saying the high dimensional and nonlinear systems are the edge cases, which isn't what's intended. So I'd suggest changing it to something like "... If we ignore the (edge) cases in which two or more terms exactly cancel..."

--

"In short, the sparse coupling conjecture says that any two states are conditionally independent if one state does not influence the other." - this should probably be qualified, e.g. "In short, the sparse coupling conjecture says that for sufficiently high-dimensional and nonlinear systems, any two states..."

--

On line 649 the conjecture is referred to as a result.

--

The text "Error! Reference source not found" appears on line 693. (It probably refers to the conjecture.)
